# Stearoyl-CoA Desaturase inhibition reverses immune, synaptic and cognitive impairments in an Alzheimer's disease mouse model

Laura K. Hamilton[1,2], Gaël Moquin-Beaudry[1,2,6], Chenicka L. Mangahas[1,2,6], Federico Pratesi[1,2,3,4], Myriam Aubin[3,4], Anne Aumont[1,2,3,4], Sandra E. Joppé[1,2], Alexandre Légiot[5], Annick Vachon[3,4], Mélanie Plourde[3,4], Catherine Mounier[5], Martine Tétreault[1,2] & Karl J. L. Fernandes[1,2,3,4 ✉]

The defining features of Alzheimer's disease (AD) include alterations in protein aggregation, immunity, lipid metabolism, synapses, and learning and memory. Of these, lipid abnormalities are the least understood. Here, we investigate the role of Stearoyl-CoA desaturase (SCD), a crucial regulator of fatty acid desaturation, in AD pathogenesis. We show that inhibiting brain SCD activity for 1-month in the 3xTg mouse model of AD alters core AD-related transcriptomic pathways in the hippocampus, and that it concomitantly restores essential components of hippocampal function, including dendritic spines and structure, immediate-early gene expression, and learning and memory itself. Moreover, SCD inhibition dampens activation of microglia, key mediators of spine loss during AD and the main immune cells of the brain. These data reveal that brain fatty acid metabolism links AD genes to downstream immune, synaptic, and functional impairments, identifying SCD as a potential target for AD treatment.

---

[1] Research Center of the University of Montreal Hospital (CRCHUM), Université de Montréal, Montreal, Canada. [2] Department of Neurosciences, Faculty of Medicine, Université de Montréal, Montreal, Canada. [3] Research Center on Aging, CIUSSS de l'Estrie-CHUS, Sherbrooke, Canada. [4] Department of Medicine, Faculty of Medicine and Health Sciences, Université de Sherbrooke, Sherbrooke, Canada. [5] Department of Biological Sciences, Faculty of Science, Université de Québec à Montréal (UQAM), Montreal, Canada. [6] These authors contributed equally: Gaël Moquin-Beaudry, Chenicka L. Mangahas. ✉email: karl.fernandes@USherbrooke.ca

Alzheimer's disease presents in familial and sporadic forms. Familial AD is caused by autosomal dominant mutations that alter processing of the amyloid precursor protein. In contrast, while there is no single cause of sporadic AD, its incidence is increased by gene variants implicated in lipid metabolism, immunity and synaptic function. Despite their distinct genetic basis, familial and sporadic forms develop similar cognitive deficits and virtually indistinguishable neuropathology, including abnormalities in amyloid, tau, lipids, immunity, and synapses. The precise links between these pathologies and their respective roles in the development of AD remain unclear.

Lipid abnormalities have been the least investigated and understood of these pathologies. Lipids are a fundamental class of biomolecules comprising thousands of individual species, and their appropriate metabolism (uptake, synthesis and/or modification) is crucial for providing cellular energy substrates, bioactive molecules and building blocks of cellular structure. Alterations in lipid species are observed in both familial and sporadic AD[1–4]. Along these lines, the single greatest genetic risk factor for sporadic AD is the ApoE4 variant, an apolipoprotein that transports fatty acids and cholesterol[5,6], and ApoE4 likewise accelerates the development of familial AD[7–9]. Such findings support the idea that lipid metabolism disturbances are a common mechanism in all forms of AD.

Stearoyl-CoA desaturase (SCD) is the rate-limiting enzyme in the conversion of saturated fatty acids to delta-9 mono-unsaturated fatty acids (MUFA), such as oleic acid[10]. Intriguingly, studies on large AD cohorts have shown increased MUFA levels in the plasma[2], and both MUFA and SCD expression are increased in the brains of AD patients and negatively correlate with cognitive status[11–13]. In a previous study, we identified periventricular lipid deposits in post-mortem AD brains and in an AD mouse model that, at least in mice, contained MUFA-rich triglycerides[3]. Intriguingly, infusion of an SCD inhibitor (SCDi) into the ventricles of pre-symptomatic AD mice reduced accumulation of MUFA-rich triglycerides and rescued the early decline of periventricular and hippocampal neural stem cell activity[3]. Since the hippocampus is a key locus of dysfunction in AD, here we investigated whether administering SCDi to symptomatic AD mice can improve hippocampal function. We found that a 1-month SCDi infusion rescued learning and memory deficits to wildtype levels and had widespread effects on immune cell activation and synaptic defects, three core features of AD. We believe these findings have important clinical implications for AD, since SCD inhibitors are currently in clinical trials for obesity and Parkinson's disease.

## Results

**The 3xTg hippocampus exhibits transcriptomic changes in MUFA metabolism and in the central pathways of sporadic AD: lipids, immunity and synapses.** The 3xTg model of AD[14] carries the human APP_{Swe}, tau_{P301L}, and PS1_{M146V} mutations and develops symptoms of learning and memory impairments as early as 6 months of age, prior to onset of overt amyloid plaques and neurofibrillary tangles[14,15]. To better understand the genetic state of the early symptomatic 3xTg hippocampus, we performed whole-hippocampus RNA-sequencing on 8-month-old female WT and 3xTg mice. This revealed 1114 differentially expressed genes (DEGs) (Fig. 1a, b), with 632 down-regulated and 482 up-regulated DEGs (Fig. 1c). Gene Ontology (GO) analysis showed that the up-regulated DEGs were enriched in GO Biological Process gene sets relating to cellular responses, immune system and protein metabolism (Fig. 1d). The down-regulated DEGs were enriched in GO Biological Process gene sets relating to

nervous system development, neuronal differentiation and morphogenesis, and cell and synaptic adhesion (Fig. 1e). The down-regulated DEGs were also enriched in GO Molecular Function gene sets that included membrane channel activity, ion binding, and lipid metabolism. Interestingly, the lipid metabolism gene sets included MUFA metabolism and SCD activity specifically (stearoyl-CoA 9-desaturase activity, $p = 0.031$; Acyl-CoA desaturase activity, $p = 0.036$; palmitoyl-CoA 9-desaturase activity, $p = 0.009$; Medium-chain fatty acid-CoA ligase activity, $p = 0.036$) (Fig. 1f and Supplemental Data File 1).

The above GO enrichments highlight lipid metabolism, immunity, and synaptic function, the same processes that encompass AD risk genes: lipid metabolism (APOE, CLU, ABCA7, ECHDC3), immune response (TREM2, CR1, MS4A, CD33, EPHA1, C4b, H2-Eb1/2, SCIMP, INPP5d), and synaptic function (PICALM, CD2AP, BIN1, CNTNAP2)[16–18]. To identify the lipid, immune, and synapse genes in our DEG list, we used gene set enrichment analysis (GSEA) and cross-referenced the enriched gene sets with lipid, immune and synapse keywords (see the "Methods" section and Supplemental Data File 1). This revealed that 177/1114 (16%) of DEGs were related to lipid metabolism (Fig. 1g), 287/1114 (26%) to the immune response (Fig. 1h) and 206/1114 (18%) to synaptic function (Fig. 1i). Notably, the list of significant DEGs included *ApoE* and *Trem2* (triggering receptor expressed on myeloid cells 2), the two genes that most increase sporadic AD risk, as well as the minor AD risk genes, *Cntnap2* (contactin associated protein 2) and *C4b* (complement component 4B) (Supplemental Data File 1). Thus, key genetic pathways associated with sporadic AD are dysregulated in the 3xTg hippocampus.

While bulk RNA sequencing cannot distinguish between differences in cell type abundance versus function, this analysis allowed us to draw three main conclusions: (1) the 3xTg model of familial AD develops changes in lipid, immune and synapse genes, biological categories implicated in the pathogenesis of sporadic AD; (2) transcriptomic changes in lipid metabolism genes are a prominent feature in the 3xTg hippocampus; and (3) defects in MUFA metabolism may specifically be involved (Fig. 1f).

**Central administration of SCDi reverses transcriptomic changes in the 3xTg hippocampus, with main effects on immune and synapse genes.** We previously reported that SCD inhibition in pre-symptomatic 3xTg mice reverses the early decline in adult neural stem cell proliferation and neuroblast formation[3]. Given that fatty acid metabolism genes, including SCD, are prominently deregulated in the hippocampus at symptomatic stages (Fig. 1), we asked whether SCD activity contributes to the observed transcriptomic disturbances.

9-month-old female 3xTg and WT mice were infused with a commercially available SCDi, ab142089, or its DMSO-containing vehicle into the lateral ventricles for 1 month using intracerebroventricular osmotic pumps. We first assessed target engagement by profiling fatty acids in the periventricular tissue via gas chromatography-flame ion detection, which confirmed an SCDi-mediated increase in levels of the main SCD substrates, C16:0 (palmitic acid) and C18:0 (stearic acid) (Supplemental Fig. 1a). We then performed bulk RNA sequencing on the hippocampi of these mice (Fig. 2a). Principle component analysis (PCA) using the entire transcriptome (all genes) showed that the mice clustered according to their experimental groups, that SCDi had a larger impact on overall gene expression in 3xTg than WT mice, and that SCDi-infused WT and 3xTg mice both remained transcriptomically distinct from the DMSO-infused WT mice (Fig. 2b). Consistent with these observations, SCDi treatment

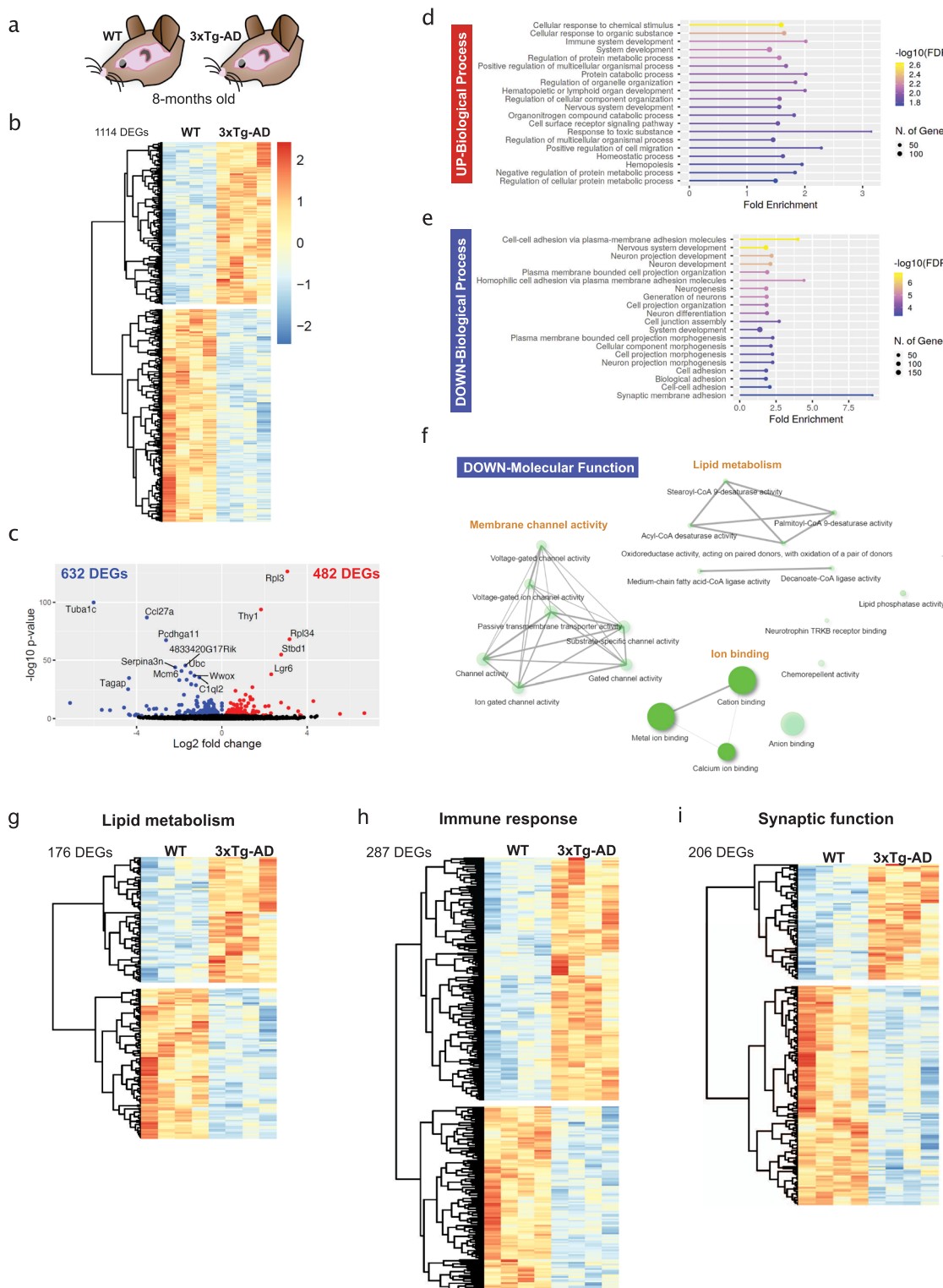

**Fig. 1 Transcriptomic changes affecting the core AD processes of lipid metabolism, immune response, and synaptic function in the 3xTg hippocampus.**
**a** Experimental paradigm for the whole-hippocampus bulk RNA sequencing experiment. **b** Heatmap of the 1114 Differentially Expressed Genes (DEGs, $p \leq 0.01$) between 8-month-old WT and 3xTg hippocampus. **c** Volcano plot of the 632 downregulated DEGs (blue) and 482 up-regulated DEGs (red). **d**–**f** GO enrichment analysis showing the top 20 most enriched GO Biological Process gene sets (**d**, **e**) and the main GO molecular function nodes (**f**). The most significant (FDR $\leq 0.05$) GO biological process gene sets for up-regulated DEGs were related to cellular responses to stimuli and the immune response (**d**) while those for the down-regulated DEGs were related to nervous system development and synapses (**e**). GO molecular function gene sets were only significantly enriched for down-regulated DEGs (FDR $\leq 0.1$) and are illustrated by network plots that highlight lipid metabolism, membrane channel activity and ion binding (**f**). **g**–**i** Heatmaps of the DEGs that are related to lipid metabolism (**g**), immune response (**h**), and synaptic function (**i**) (see the "Methods" section). Refer to Supplemental Data File 1 for the complete DEG list and the lipid metabolism, immune response, synaptic function genes.

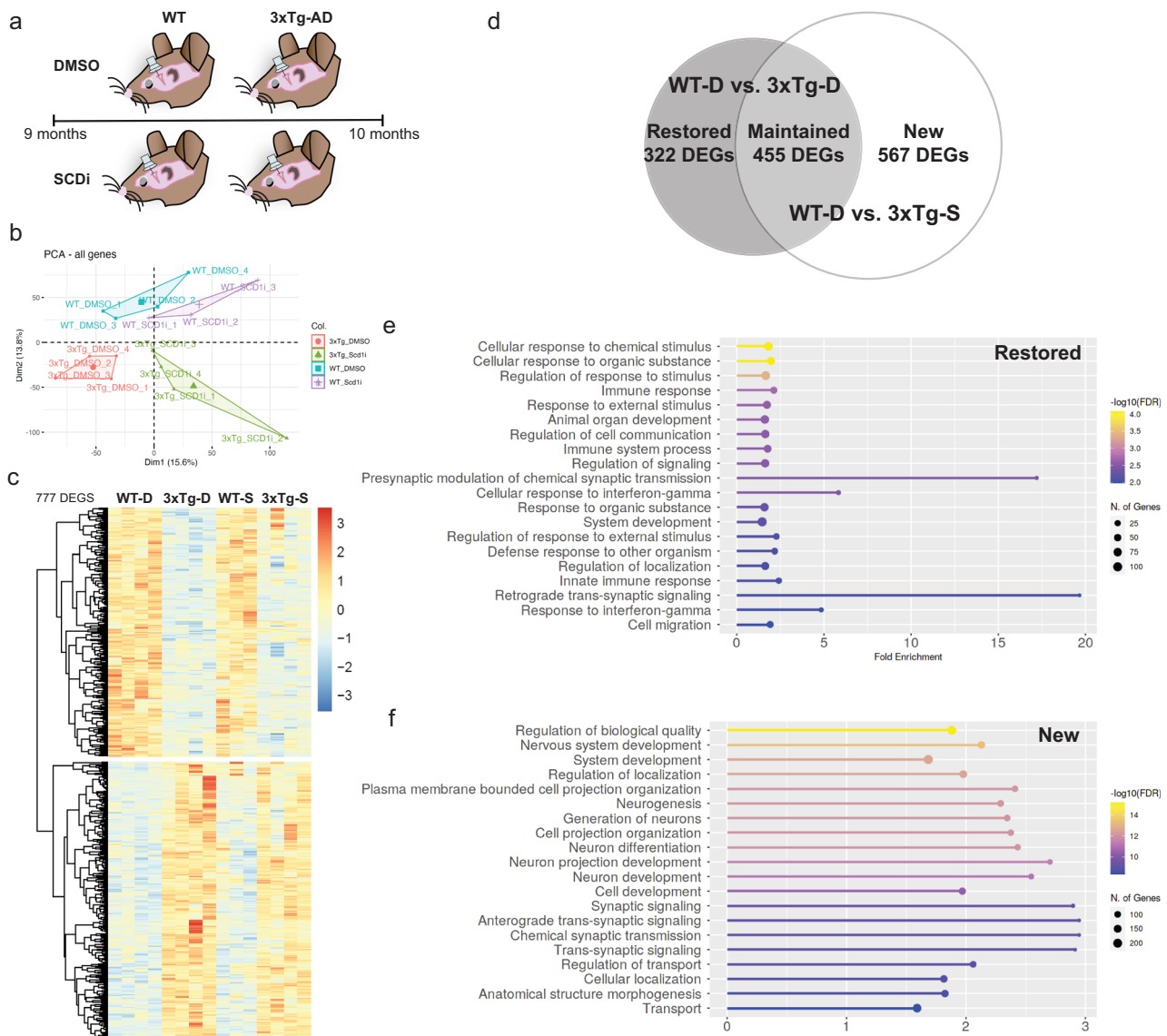

**Fig. 2 SCDi infusion potently modulates gene expression in the 3xTg hippocampus, with main effects on immune- and synapse- related genes.**
**a** Timeline of intracerebroventricular infusion experiment on the 9-month-old WT and 3xTg mice infused with either vehicle (DMSO) or SCD inhibitor (SCDi). Hippocampi were extracted after 1 month infusion and processed for whole hippocampus bulk RNA sequencing. **b** Principal component analysis of the four experimental groups using the whole transcriptome ("all genes"). **c** Heatmaps of the 777 DEGs ($p \leq 0.01$) between WT-D and 3xTg-D groups, with DEG expression shown across the four treatment groups ($n = 4$ mice/group). **d** Venn diagram showing the overlap of the WT-D/3xTg-D and WT-D/ 3xTg-S DEG lists, in order to identify DEGs that are "Restored" (no longer significant), "Maintained" (still significant), or "New" (newly appearing) after SCDi infusion into 3xTg mice. **e**, **f** GO enrichment analysis showing the top 20 most enriched GO biological process gene sets for the "Restored" DEGs (**e**) and "New" DEGs (**f**) in SCDi-infused 3xTg mice. Restored gene were mainly enriched in GO Biological Process gene sets related to cellular response to stimuli, immunity and synapses (**e**) while the newly changed genes were mainly enriched in GO biological process gene sets related to nervous system development and synapses (**f**). See Supplemental Data File 2 for complete DEG lists.

resulted in more statistically significant DEGs in 3xTg mice (3xTg-D vs. 3xTg-S, 434 DEGs, Supplemental Data File 2) than in WT mice (WT-D vs. WT-S, 9 DEGs, Supplemental Data File 2), and the total number of DEGs between WT-D and 3xTg-D groups (777 DEGs) and between WT-D and 3xTg-S groups (1022 DEGs) were similarly high.

To better understand the effects of SCDi on the transcriptomic differences between WT and 3xTg mice, we compared the WT-D/3xTg-D DEG list with the WT-D/3xTg-S DEG list (Fig. 2c, d). 455/777 of the WT-D/3xTg-D DEGs still remained after SCDi infusion (WT-D versus 3xTg-S); these DEGs were annotated as "Maintained" (Fig. 2d). Notably, 322/777 of the WT-D/3xTg-D DEGs were no longer significant in the WT-D/3xTg-S

comparison; expression of these DEGs were reduced by an average of 42% (Supplemental Fig. 1b) and were annotated as "Restored" (Fig. 2d). 567 DEGs in the WT-D/3xTg-S comparison were not present in the WT-D/3xTg-D comparison; we annotated these as "New" (Fig. 2d). GO analysis showed that "Restored" genes were enriched in GO biological processes relating to cellular responses to various stimuli, immune system, and synapses (Fig. 2e), while "New" DEGs showed notable enrichments for nervous system development and synapse-related processes (Fig. 2f). No lipid-related GO categories were significantly enriched in either the Restored or New gene lists, but lipid-related DEGs represented 13% of Restored DEGs and 20% of New DEGs (Supplemental Data File 2).

We confirmed and validated the specificity of this SCDi, ab142089, on 3xTg gene expression by using CAY10566 (a second commercially available SCDi that has distinct structural characteristics). Bulk RNA sequencing analysis on a new cohort of 9-month-old mice infused with CAY10566 (Supplemental Fig. 2a) revealed that the number of "Restored", "Maintained" and "New" DEGs was virtually identical as in our original experiment with ab142089 (Supplemental Fig. 2b, c). GO analysis showed that, like ab142089, the Restored and New DEGs for CAY10566 were enriched in GO Biological Process gene sets related to synapses and nervous system growth/development (Supplemental Fig. 2d, e). Overlaying the ab142089 New and Restored DEGs lists with the same genes from the CAY10566 experiment revealed a strong overlap for the Restored genes (Supplemental Fig. 2f). Thus, the Restored GO Biological Processes in particular may be important in mediating the drug effects.

Together, these data show that ICV administration of SCDi for 1 month profoundly impacts the 3xTg hippocampal gene expression profile. SCD inhibition restores over 40% of the DEGs between WT and 3xTg mice and this effect is generalizable to multiple SCDi. Importantly, insight provided by GO enrichment analysis suggests that SCD inhibition may be impacting biological processes related to nervous system development and synapses.

**SCD inhibition reverses 3xTg deficits in spatial learning and memory**. Given the effects of SCD inhibition on the hippocampal transcriptome, we tested whether 3xTg deficits in hippocampus-regulated cognitive functions are beneficially impacted by SCDi administration. 9-month-old WT and 3xTg mice were again infused with SCDi or vehicle for 1 month and then tested for changes in learning and memory and/or anxiety (dorsal and ventral hippocampal function, respectively).

Learning and memory were assessed using the Morris water maze (MWM). Mice were trained to learn a hidden platform location in the MWM for 4 consecutive days (6 trials/day); learning was defined as a decreasing latency to find the platform over the training period, while memory impairment was defined by reduced platform crossings during a probe trial 24 h after the last training session. Learning curves showed that both WT and 3xTg mice learned the platform location over time, with 3xTg-D and 3xTg-S mice, respectively, showing longer latencies to escape than their WT-D and WT-S counterparts (2-way ANOVA, repeated measures; time factor, $p < 0.0001$; strain factor, $p = 0.0242$; no interaction) (Fig. 3a). While the latency did not differ between experimental groups on individual training days, cumulative latency (area under the curve, AUC) showed that the 3xTg-D group had a learning deficit (two-way ANOVA, Dunnett's post-hoc, $p = 0.0177$) and this deficit was corrected by SCDi (WT-D/3xTg-S, Dunnett's post-hoc, $p = 0.4729$) (Fig. 3a). Platform crossings on the Day 5 probe test showed that the 3xTg-D group exhibited a memory impairment (two-way ANOVA, Dunnett's post-hoc $p = 0.04$) that was likewise reversed by SCDi (WT-D/3xTg-S, Dunnett's post-hoc, $p = 0.61$) (Fig. 3b). SCDi-mediated improvements in MWM performance were not attributable to changes in swim speed, which was higher in 3xTg than WT mice and unchanged by SCDi (WT-D, 14.3 cm/s; 3xTg-D, 18.8 cm/s; WT-S, 15.9 cm/s; 3xTg-S, 18.8 cm/s; two-way ANOVA, strain factor $F(1,16) = 10.72$, $p = 0.005$). In contrast to the effects on learning and memory, SCDi-mediated improvements were not observed for anxiety, which was assessed using the light-dark box (Supplemental Fig. 3a, b), open field test (Supplemental Fig. 3c), and elevated plus maze (Supplemental Fig. 3d–f). Although 3xTg mice exhibited greater baseline anxiety than WT mice in each of these tests, no evidence of SCDi-induced changes in anxiety was

detectable. Thus, 1-month administration of SCDi selectively reverses the dorsal hippocampus-associated deficits in spatial learning and memory seen in symptomatic 3xTg mice.

Interestingly, unlike its effects in pre-symptomatic 3xTg mice (2–3 months old),[3] SCDi infusion in these 9-month-old symptomatic mice did not rescue neural stem/progenitor proliferation or neuroblast numbers in either the hippocampal dentate gyrus (Fig. 3c–f) or the forebrain subventricular zone (Fig. 3g–j). We also assessed whether SCDi affected AD hallmarks such as amyloid, tau or neuronal loss. While 3xTg mice had significantly higher loads of amyloid (both total and oligomeric) and hyperphosphorylated Tau, SCDi treatment did not alter these pathologies (Supplemental Fig. 4a–i). We also did not detect treatment-induced changes in neuronal density or layer thickness in the DG (Fig. 3k, l) or CA1 (Fig. 3m, n).

Overall, these data show that SCDi is sufficient to rescue learning and memory impairments in 3xTg mice, while anxiety, neural stem/progenitor proliferation, amyloid load, tau hyperphosphorylation and neuronal loss remain unchanged.

**SCDi rescues hippocampal spine number and dendritic complexity**. Synaptic loss is prominent in post-mortem brains of AD patients[19] and AD transgenic animal models[20,21] and is considered the strongest correlate of cognitive deficits[22,23]. Since 18% of DEGs between WT and 3xTg hippocampus were synapse-related (Fig. 1i), we extended our bioinformatic analyses to quantify the synapse-related DEGs that were restored, maintained or newly changed upon SCDi infusion (Supplemental Data File 2). This revealed that SCDi "restored" 39% of synapse DEGs (56/142 genes) and changed expression of an additional 142 "new" synapse genes (Supplemental Data File 2a–d), suggesting a potential effect on hippocampal synaptic connections.

To test if SCDi affects spine number and dendritic structure, we treated WT and 3xTg mice for 1 month with SCDi or DMSO vehicle as previously and used Golgi-Cox staining to quantify dendritic spine numbers, dendritic length, and dendritic complexity in the dentate gyrus (DG) and the CA1 regions of the dorsal hippocampus. In 3xTg-D mice, spine densities were significantly reduced on secondary and tertiary branches of both DG granule neurons (Fig. 4a, b) and CA1 pyramidal neurons (Fig. 4c, d). Remarkably, in both these hippocampal subfields, SCDi rescued spine density to WT-D levels (Fig. 4a–d). Dendritic length in 3xTg-D mice was reduced in the CA1 (Fig. 4g), but not the DG (Fig. 4e), and rescued to WT-D levels by SCDi (Fig. 4g). Similarly, Scholl analysis of dendritic arbor complexity identified deficits in the CA1 (Fig. 4h), but not the DG (Fig. 4f), of 3xTg-D mice and SCDi rescued the CA1 impairment (Fig. 4h). To confirm the striking rescue of hippocampal spine loss by SCDi, we repeated our analysis in a new cohort of mice treated with a second SCDi (CAY10566) (Supplemental Fig. 2a); again, we found a complete rescue of dendritic spines in both the DG (Supplemental Fig. 5a, b) and CA1 (Supplemental Fig. 5c, d).

To explore whether the recovery of dendritic spines and dendritic structure was associated with increased markers of cellular activation within the hippocampus, we examined the expression of 18 hallmark immediate-early genes (IEGs) within our bulk RNA-sequencing dataset. IEG expression is rapidly triggered in neuronal and non-neuronal cells in response to diverse signals, and in neurons in particular, IEGs such as *Egr-1*, *c-fos*, *Arc* and *Bdnf* are dynamically modulated by depolarization[24–27]. Strikingly, 11/18 IEGs were down-regulated in 3xTg-D compared to WT-D mice (2-way ANOVA, Dunnett's post-hoc test, $p = 0.001$), and after SCDi infusion, 3xTg-S mice no longer differed from WT-D mice (Dunnett's post-hoc test, $p = 0.937$) (Fig. 4j).

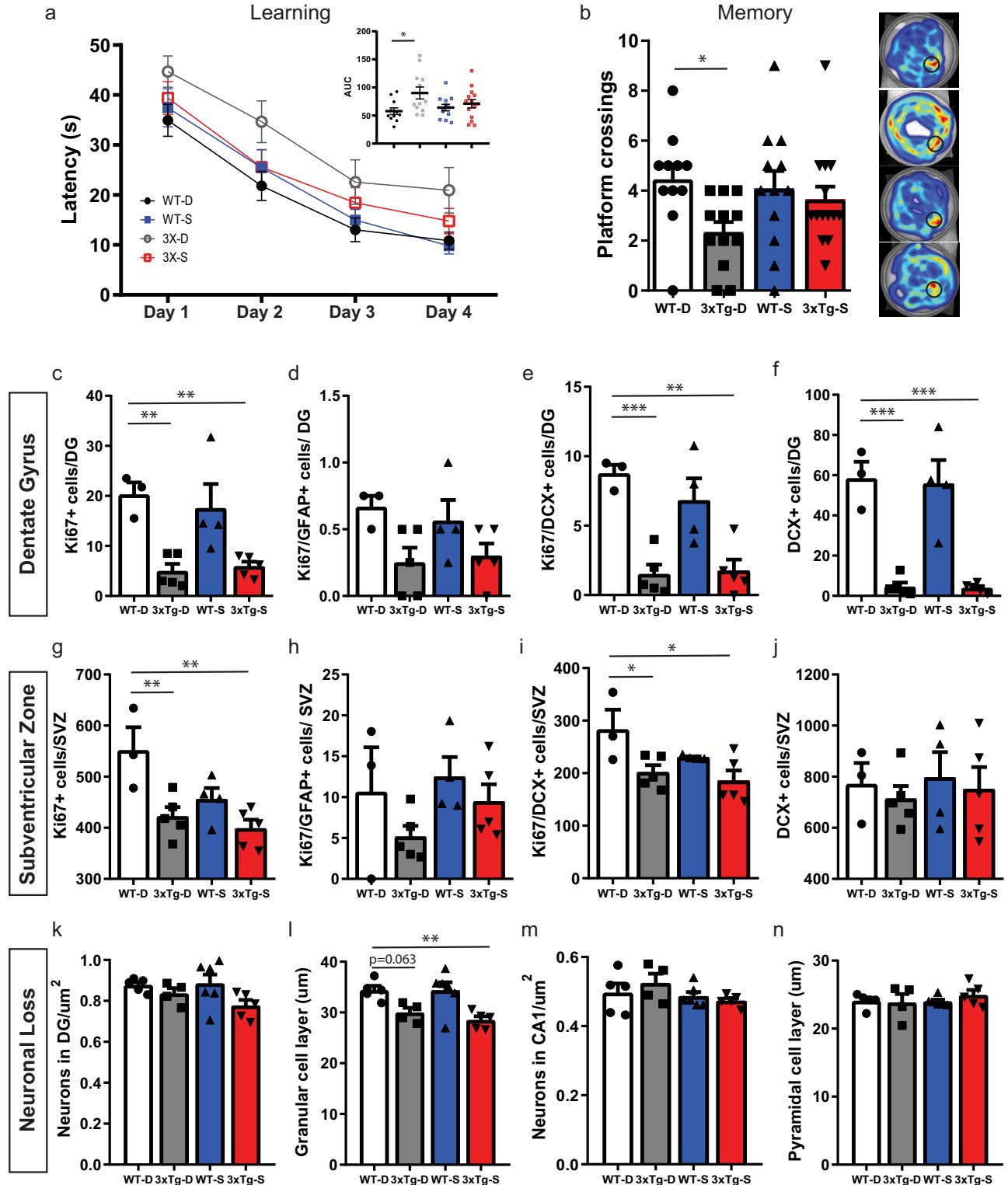

Overall, these experiments show that SCDi administration in 3xTg mice results in broad effects on hippocampal synapse-related gene expression and leads to recovery of hippocampal dendritic spines, dendritic structure and activity-related IEG expression.

**Single-cell analysis identifies MHC-I genes as main targets of SCDi in 3xTg hippocampal microglia.** Microglia, the resident immune cells of the central nervous system, are crucial for normal synapse development, support and elimination[28,29]. They also disproportionately express AD risk genes[30] and are implicated in AD-associated synapse loss[29,31], making them central players during AD pathogenesis[32,33]. Given that SCDi "restored" 39% of immune DEGs (94/238) and stimulated 159 "new" immune-related DEGs in the 3xTg hippocampus (Supplemental Data File 2), we asked whether SCDi-mediated changes in microglial activity contributes to its beneficial effects on dendritic spines.

**Fig. 3 SCDi infusion in symptomatic 3xTg mice improves spatial learning and memory without modulating neural stem/progenitor activity or neuronal loss. a, b** Learning and memory in the Morris water maze (WT-D, $n = 11$; 3xTg-D, $n = 12$; WT-S, $n = 12$; 3xTg-S, $n = 14$). Learning (latency) average of 6 trials/day/mouse (**a**). Two-way repeated measures ANOVA identified significant time ($F(2.539, 114.3) = 77.88$, $p < 0.0001$) and strain ($F(3,45) = 3.452$, $p = 0.0242$) effects, with no interaction. Area under the curve (AUC) (inset) showing higher cumulative latency in 3xTg-D group that is not present in the 3xTg-S group (two-way ANOVA, strain effect $F(1,45) = 6.363$, $p = 0.0153$, strain/treatment interaction of $F(1,45) = 2.634$, $p = 0.1116$), Dunnett's post-hoc test, $p = 0.0177$). Memory (platform crossings) (**b**). Two-way ANOVA showed strain effect ($F(1,45) = 5.114$, $p = 0.0286$) and that the 3xTg-D group, but not the WT-S or 3xTg-S groups, was significantly different from the WT-D group (Dunnett's post-hoc test, $p = 0.0351$). Representative heatmaps of swim trajectories (right). Error bars represent mean ± SEM. *$p \leq 0.05$. **c–j** Quantification of proliferation and neuroblasts in the DG (**c–f**) and SVZ (**g–j**) of 10-month-old WT-D ($n = 3$), 3xTg-D ($n = 5$), WT-S ($n = 4$), and 3xTg-S mice ($n = 5$). We observed decreased total proliferating cells in the DG (WT-D/3xTg-D, $p = 0.005$, WT-D/3xTg-S, $p = 0.008$) (**c**) and in the SVZ (WT-D/3xTg-D $p = 0.008$, WT-D/3xTg-S, $p = 0.002$) (**g**). No changes were detected in Ki67+/GFAP+ proliferating neural stem cells in the DG (**d**) or SVZ (**h**). We found fewer Ki67+/DCX+ proliferating neuroblasts in the DG (WT-D/3xTg-D, $p = 0.0009$, WT-D/3xTg-S, $p = 0.0012$) (**e**) and in the SVZ (WT-D/3xTg-D, $p = 0.03$, WT-D/3xTg-S, $p = 0.01$) (**i**) as well as a decrease in DCX + total neuroblasts in the DG (WT-D/3xTg-D, $p = 0.002$, WT-D/3xTg-S, $p = 0.0002$) (**f**) with no difference in the SVZ (**j**). No SCDi-mediated change was observed in any of these parameters of neural stem/progenitor activity (two-way ANOVA, Dunnett's post-hoc test). DG Dentate gyrus, SVZ subventricular zone. Error bars represent mean ± SEM. *$p \leq 0.05$, **$p \leq 0.01$, ***$p \leq 0.001$. **k–n** Quantification of Nissl-stained neuron number (**k**) and granular layer thickness (**l**) in the DG and neuron number (**m**) and pyramidal layer thickness in the CA1 (**n**) of WT-D ($n = 5$), 3xTg-D ($n = 4$), WT-S ($n = 6$), 3xTg-S ($n = 5$) mice. Note no SCDi-mediated change was observed in neuronal loss parameters (two-way ANOVA, Dunnett's post-hoc test). Error bars represent mean ± SEM. **$p \leq 0.01$. Source data are provided as a Source Data file.

---

In line with this idea, immunostaining of Iba1+ microglia and PSD95+ postsynaptic densities intriguingly revealed a decreased colocalization in the hippocampal CA1 of 3xTg-D mice (2-way ANOVA, Dunnett's post-hoc test, $p = 0.012$) that was restored to WT-D levels by SCDi (2-way ANOVA, Dunnett's post-hoc test, $p = 0.963$) (Fig. 5a, b).

Microglia are challenging to study in their native environment as they comprise only 5–10% of hippocampal cells, exist in multiple cell states, and are hyper-sensitive to tissue processing stress. We established an optimized single-cell RNA sequencing strategy in which stressful purification steps were removed from the cell dissociation protocol and the gentle, microwell-based BD Rhapsody single cell system was used to maximize cell viability and yield. We also used a CD45-based multiplexing protocol to efficiently tag all immune cells and eliminate batch effects by allowing all experimental groups to be processed in the same sequencing run (see the "Methods" section).

Using the above strategy on WT and 3xTg mice infused with SCDi or vehicle as previously, 7357 CD45-tagged cells were recovered from 17,893 hippocampal cells that had met quality control thresholds (Supplemental Fig. 6). Sequencing depth reached a mean of 31,681 reads/cell. UMAP dimension reduction methods identified 33 cell clusters (Supplemental Fig. 6e) whose cell type identities were attributed using previously published gene sets[34–37] (Fig. 5c, Supplemental Fig. 6f). Virtually all the recovered immune cells were successfully CD45-tagged, including 99% of microglia, 100% of T-cells, 93% of B-cells, and 97% of macrophages. As expected, other main clusters of recovered brain cells were present but incompletely tagged, including astrocytes, oligodendrocytes and endothelial cells (Supplemental Fig. 6a, b); to avoid subpopulation biases, these non-immune cells were not further studied.

Microglia were by far the largest cluster of CD45-tagged hippocampal cells (~50%) and their total number was higher in 3xTg mice regardless of treatment (Fig. 5d). Bioinformatic comparison of the microglia clusters in the WT-D and 3xTg-D groups revealed 122 microglial DEGs (Fig. 5f). Overlapping these 122 WT-D/3xTg-D DEGs with the WT-D/3xTg-S DEGs showed that 24% (30/122) were restored (Fig. 5f, g) and 34 genes were newly changed (Fig. 5f, h) by SCDi. Intriguingly, direct comparison of microglia in the 3xTg-D and 3xTg-S groups identified only five genes that were significantly modulated by SCDi in the 3xTg strain, and all five of these were MHC class I (MHC-I) genes: *H2-D1*, *H2-K1*, *H2-T23*, *H2-Q6*, and *H2-M3* (Fig. 5i). Further analysis revealed that *H2-D1* and *H2-K1* were expressed in over ~75% of microglia while *H2-T23*, *H2-Q6*, and *H2-M3* were only expressed in ~25%

(Fig. 5i), and that all were upregulated within 3xTg-D microglia and downregulated in 3xTg-S microglia (Fig. 5i). Notably, both *H2-D1* and *H2-K1* were also among the 30 "restored" genes (Fig. 5g).

MHC-I genes are relevant in the context of our earlier data on SCDi-mediated spine recovery, as MHC-I has been implicated in microglia-driven synaptic pruning[38] and learning and memory[39]. To assess if the SCDi-mediated changes in MHC-I genes are relevant to other AD mouse models, we compared our WT-D/3xTg-D microglia DEG list with DEG lists from four distinct AD models, including bulk RNA-sequencing data sets from 5xFAD[40] or APP-SAA (Xia, GSE158153) microglia and single-cell RNA sequencing from App[NL-G-F] KI mice[34] or APP/PS1[36] microglia (Fig. 5j). This analysis revealed an overlap of 12 DEGs across all five AD models (*Abcd2*, *Bank1*, *Ccl6*, *Ctsl*, *Fam102b*, *Filip1l*, *H2-D1*, *H2-K1*, *Nrp1*, *Pnp*. *Rplp1*, *Rps13*). Together, these data indicate that the targets of SCDi (*H2-D1* and *H2-K1*) are dysregulated in microglia in multiple AD mouse models.

To study the association between SCD, MHC-I and microglial activation, we cultured neonatal microglia from WT and Scd1-KO mice and measured their responses to an inflammatory stimulus, lipopolysaccharide (LPS). qPCR analysis showed that LPS treatment of WT microglia significantly increased mRNA levels of *TNFα* and *IL1β* (two markers of microglia activation) and that this upregulation was blunted in Scd1-KO microglia (Fig. 5k, l). WT microglia were likewise protected from LPS-induced inflammatory activation when treated with SCDi, expressing similar levels of *TNFα* and *IL1β* as the LPS-treated Scd1-KO microglia (Fig. 5k, l). Furthermore, expression of *H2-D1* and *H2-K1* mRNAs were increased following LPS exposure and this increase was blunted in both Scd1-KO microglia and in WT microglia treated with SCDi (Fig. 5m, n). These in vitro data identify a regulatory role of Scd during inflammatory activation of microglia and demonstrate that inflammation-induced upregulation of MHC-I genes in microglia is largely mediated by microglial Scd expression.

Collectively, these single cell and culture data reveal that Scd activity regulates microglial activation/response, and since the MHC-I genes *H2-D1* and *H2-K1* are the main transcriptional targets of SCDi, an Scd-MHC-I pathway may play an important role during this process.

**Single-cell analysis reveals that SCDi alters the cellular land-scape of microglia in the 3xTg hippocampus.** Previous single-cell RNA sequencing studies have identified several discrete

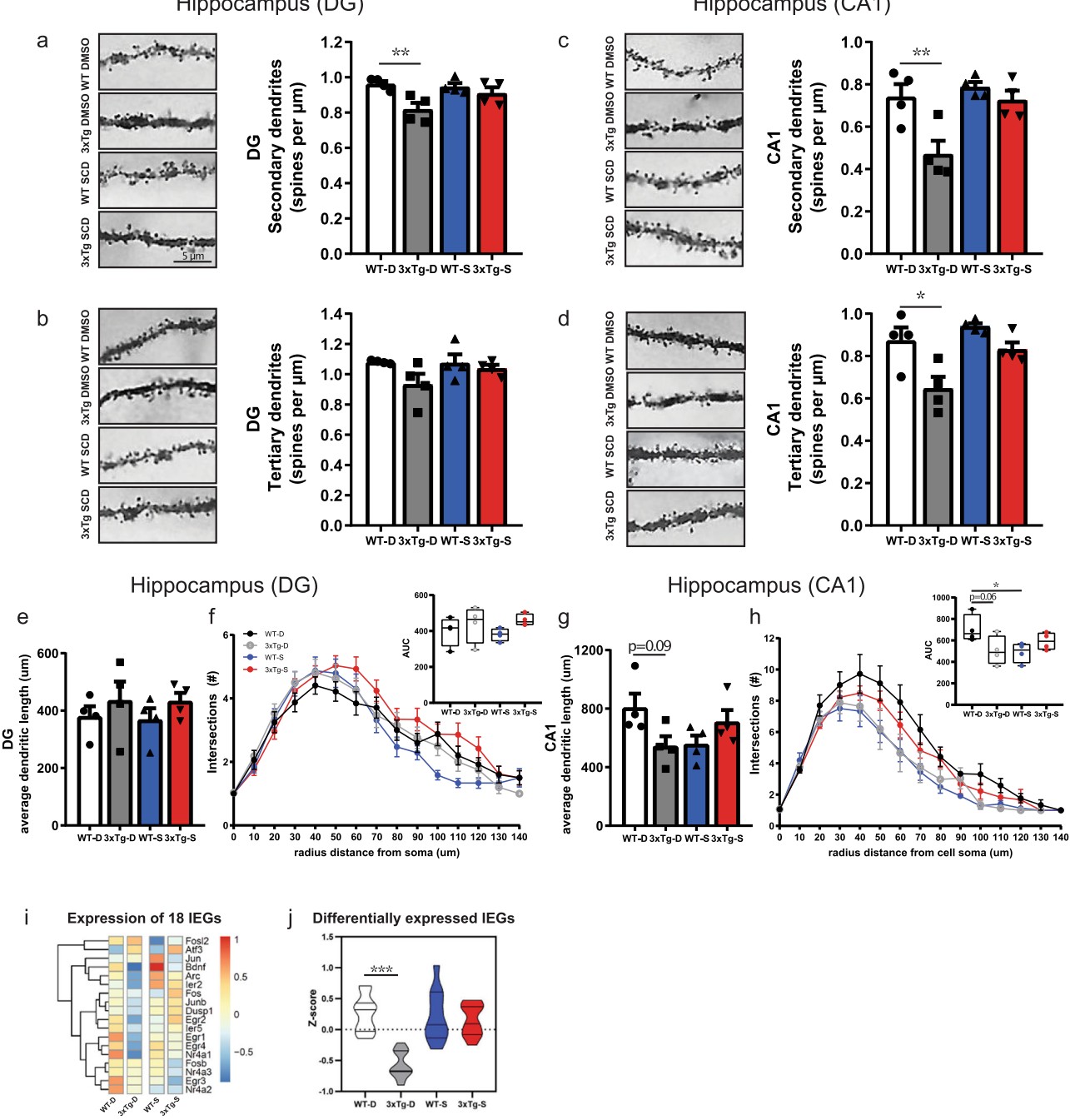

subtypes or physiological states of microglia within the brain's microglial population[34–36,40]. To characterize microglial subtypes in the 3xTg hippocampus, we subclustered our single-cell RNA sequencing microglial population using tSNE dimensionality reduction (Fig. 6a). The seven resulting subclusters all co-expressed classical microglia genes such as *P2ry12*, *Tmem119*, *Slc2a5*, and *Alf3*, as well as subcluster-specific DEGs (Fig. 6b). To identify these subclusters, we first used previously published datasets (Supplemental Fig. 7a–c)[34,35], enabling us to annotate microglial subclusters 1 and 2 as Resting/Homeostatic (RH1 and RH2), subcluster 5 as Active Response (AR) and subcluster 6 as Active Interferon Response (AIR). We then used PAGA-tree pseudotime analysis[41] to infer the lineage relationship between subclusters, revealing subclusters 3 and 4 to be likely transition states (Fig. 6c). Subcluster 3 was annotated as a transitional state linking RH1/RH2 with AIR and AR (i.e., T(AIR,AR)). Subcluster

4 was annotated as a transitional state between RH1/RH2 and subcluster 7 (Fig. 6c). Subcluster 7 expressed 311 DEGs that included high expression of phospholipid phosphatase (*Plpp3*), sodium/potassium transporting ATPases (*Atp1a2,b2*), and the Slc1a2 glutamate transporter (Fig. 6b). Like the AR and AIR microglia subtypes, subcluster 7 showed increased pseudotime transition by temporal inference (Fig. 6c) and elevated nCount RNA (Fig. 6d), suggestive of a more active state. GO analysis of subcluster 7 DEGs revealed enrichment for neuron-related bio-logical processes (Supplemental Fig. 7e), lipid and ion regulation molecular functions (Supplemental Fig. 7f) and synaptic cellular components (Supplemental Fig. 7g). Based on these features we annotated subcluster 7 as a Active Synapse-associated Response (ASR) subcluster, and the transitional subcluster 4 as T(ASR). Average mitochondrial content was similar across subclusters, ranging from 3.3% to 8.2% (Supplemental Fig. 7d), thus

**Fig. 4 SCDi infusion rescues spine number and dendritic complexity in hippocampal neurons of 3xTg mice. a–h** Golgi staining in the hippocampus. Quantification of spine number on dentate gyrus (DG) (4 animals/group, average of 6–16 neurons/animal) (**a**, **b**) or CA1 neurons (4 animals/group, average of 6–14 neurons/animal) (**c**, **d**). In the DG, secondary dendrites were significantly decreased between WT-D/3xTg-D ($p = 0.001$) but not following SCDi treatment, WT-D/3xTg-S ($p = 0.450$) (**a**), and tertiary dendrites showed a trend towards a decrease between WT-D/3xTg-D ($p = 0.111$) but not following SCDi treatment, WT-D/3xTg-S ($p = 0.857$) (**b**). In CA1, secondary dendrites were significantly decreased between WT-D/3xTg-D ($p = 0.007$) but not following SCDi treatment, WT-D/3xTg-S ($p = 0.993$) (**c**), and tertiary dendrites were also significantly decreased between WT-D/3xTg-D ($p = 0.011$) but not following SCDi treatment, WT-D/3xTg-S ($p = 0.847$) (**d**). Two-way ANOVA, Dunnett's post-hoc test. Quantification of the total dendritic length (**e**, **g**) or dendritic complexity (**f**, **h**) of neurons in the DG (4 animals per group, average of 6–16 neurons/animal) (**e**) or CA1 (4 animals per group, average of 4–13 neurons/animal) (**g**). Note a trend towards shorter dendrites between WT-D/3xTg-D ($p = 0.086$) that was not observed following SCDi treatment, WT-D/3xTg-S ($p = 0.721$). Sholl analysis of dendritic complexity in the DG (**f**) or CA1 (**h**) with box and whisker plots showing minimum, 1st quartile, median, 3rd quartile and maximum of area under the curve (AUC) (inset). Note a trend towards lower AUC between WT-D/3xTg-D ($p = 0.0613$) and not following SCDi treatment, WT-D/3xTg-S ($p = 0.369$) and a significant difference between WT-D/WT-S ($p = 0.042$). Two-way ANOVA, Dunnett's post-hoc test. *$p \leq 0.05$, **$p \leq 0.01$. Error bars represent mean ± SEM. Scale bar, 5 μm. SCDi: Stearoyl-CoA desaturase inhibitor. **i–j** Immediate-early gene (IEG) expression in the hippocampus. Heatmap of the z-score of 18 IEGs (**i**). Violin plot of the 11 IEGs that are down-regulated between WT-D and 3xTg-D; note that expression of these 11 IEGs is significantly decreased between WT-D/3xTg-D ($p = 0.0006$) and upregulated to WT-D levels in 3xTg-S mice (WT-D/3xTg-S, $p = 0.937$), two-way ANOVA, Dunnett's post-hoc test. (**j**). Violin plots represent minimum, 1st quartile, median, 3rd quartile and maximum. Error bars represent mean ± SEM. ***$p \leq 0.001$ See Supplemental Data File 2 for synapse related DEG lists. Source data are provided as a Source Data file.

subclassification of microglia is unlikely to be driven by differences in microglial quality.

We then asked whether the landscape of microglial states in the 3xTg hippocampus is altered by SCDi administration. In the WT-D control group, 43% of microglia were resting (RH1/RH2), 31% were within the AR/AIR lineage (T(AR/AIR), 27%; AR, 2%; AIR, 2%) and 26% were within the ASR lineage (T(ASR), 19%; ASR, 7%). Comparison of these microglial subclusters across treatment groups revealed that 3xTg-D mice have an expansion in their proportion of microglia within the AR/AIR lineage, and that this expansion was reversed by SCDi (Fig. 6e). Specifically, resting microglia (RH1/RH2) decreased from 43% to 36% in 3xTg-D mice while their AR/AIR lineage showed an increase from 31% to 40%, including a more than doubling of the combined downstream AR and AIR subclusters (from 4% to 10%). In 3xTg-S mice, these changes were largely corrected, with the AR/AIR lineage cells decreasing from 40% to 33%) and the RH1 cells increasing from 25% to 32% (Fig. 6e). The lineage-specific activation score showed that 3xTg-D mice had an increased percentage of activated (AR/AIR and ASR) relative to resting (RH1/RH2) lineages and that SCDi reversed this shift (Fig. 6f).

Lastly, we plotted expression of MHC-I and Scd genes as a function of microglial subclusters. The *H2-D1* and *H2-K1* genes were expressed across all microglial clusters and were most highly expressed in the AR and AIR microglia subclusters (Fig. 6g). In 3xTg-D mice, their relative expression was increased in resting (RH1/RH2) and transitional (T(AR/AIR, T(ASR)) populations, and decreased in resting, transitional and activated subpopulations by SCDi (Fig. 6h, i and Supplemental Fig. 8a, b). The main Scd family members, *Scd1* and *Scd2*, are generally most expressed in the activated microglial lineages and increased in 3xTg mice (Supplemental Fig. 8c, d). This suggests that SCDi acts on the activated microglial lineages and reverts them towards resting states.

This single cell analysis of hippocampal microglia provides a high-resolution view of microglial subtypes/states, how they are affected in an AD-like environment, and their response to SCDi. Our data demonstrate that SCDi administration reverses the AD-associated shift from resting to activated/disease states, suggesting that an SCD-controlled balance between saturated and mono-unsaturated fatty acids is a key regulator of microglial activation. Taken together with its effects on the hippocampal transcriptome, dendritic structure, activity associated IEG expression and spatial learning and memory, these findings reveal that SCDi can correct the core immune, synaptic and behavioral defects in an animal model of AD.

## Discussion

Here, we show that microglia activation, synaptic loss and learning and memory deficits, can be restored by inhibiting a single lipid metabolism enzyme, SCD. Using the 3xTg mouse, which was designed to reproduce the classical amyloid plaque and neurofibrillary tangle AD pathologies[14], we determined that the hippocampus develops principal transcriptomic alterations in the core biological processes implicated in both familial and sporadic human AD: lipid metabolism, immune response and synaptic function[17,42]. Bioinformatic analysis revealed that molecular functions involving MUFA metabolism were specifically altered in the 3xTg hippocampus; this was compelling, as we previously showed lipid droplet accumulation along the ventricle walls of human AD patients and 3xTg mice[3]. In that prior study, we demonstrated that this lipid accumulation in 3xTg mice begins at pre-symptomatic stages, worsens with age, and consists of triglycerides rich in MUFAs. We used SCDi to reduce levels of these AD-associated triglycerides: in young adult mice this recovered neural stem/progenitor proliferation surrounding the lateral ventricles and, of relevance here, in the hippocampal dentate gyrus[3]. Given the key role of the hippocampus in AD, in this study we focused specifically on understanding how SCDi affects the biology and function of the 3xTg hippocampus. While SCDi did not restore hippocampal stem/progenitor proliferation in symptomatic 3xTg mice, we demonstrate that it (i) profoundly modulates immune and synapse genes, (ii) shifts the upregulation of activated microglia towards homeostatic/resting states, (iii) restores dendritic spines, dendritic complexity and activity-associated IEG expression, and (iv) ultimately recovers spatial learning and memory function.

New tools for localizing, identifying and quantifying lipids and the genes that regulate them have highlighted the importance of the SCD-regulated balance between saturated and unsaturated fatty acids in aging and neurodegenerative diseases. Appropriate fatty acid metabolism is important for normal aging and lifespan in both invertebrates and vertebrates[43]. In AD patients, lipidomic studies show that fatty acid desaturation is altered in the brain[11,13] and plasma[2,12]. These changes might be the direct result of altered SCD activity, as *SCD* is specifically deregulated in the brains of AD patients[11] and 3xTg mice[3]. Moreover, increased fatty acid desaturation was positively correlated with increased reversal learning errors and decreased cognitive

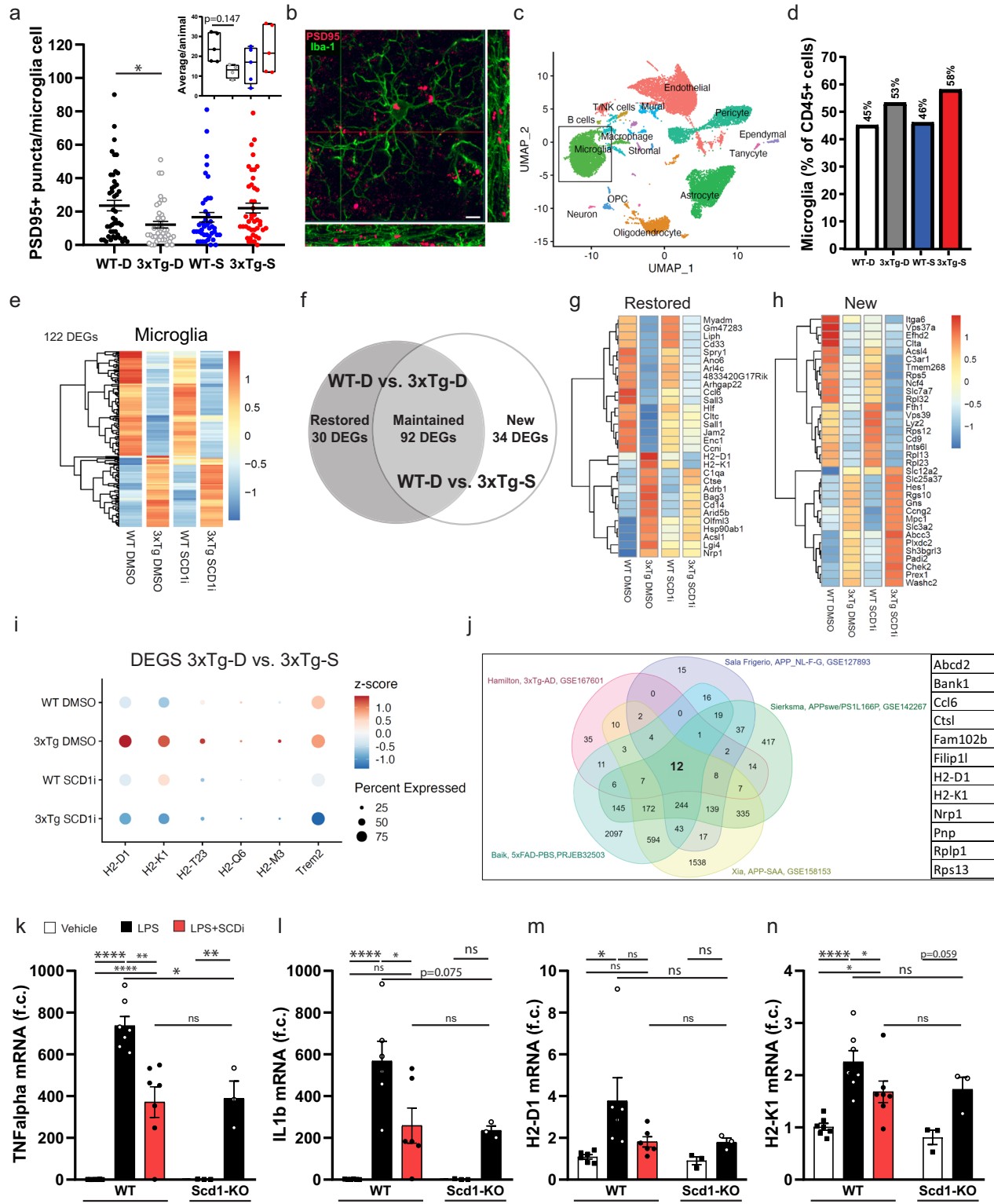

performance in aged canines[44]. Here, when we performed RNA sequencing on the whole 3xTg hippocampus, unbiased GO enrichment analysis specifically highlighted fatty acid desaturase activity as a main dysregulated molecular function. The immediate consequences of altered desaturase activity are likely to be pleiotropic, as the cellular functions of lipids as energy substrates, bioactive molecules and building blocks are all determined by their precise identity. Indeed, recent lipidomic analysis of two large human AD cohorts revealed increased

incorporation of SCD-dependent MUFAs in many complex lipid species[2]. Interestingly, beneficial effects of SCD inhibition have recently been observed in multiple animal models of Parkinson's disease[45], where SCD inhibition prevented PD-like neuropathology and motor deficits. Mechanistically SCD inhibition was shown to inhibit α-synuclein-induced lipid accumulation, toxicity, and neuronal degeneration in Parkinson's disease models[46–48]. Whether microglia play a role in the beneficial effects of SCDi was not examined in those studies.

**Fig. 5 Single-cell RNA sequencing identifies MHC-I genes as main targets of SCDi in 3xTg hippocampal microglia. a, b** Immunohistochemistry of PSD95 and Iba-1 in the hippocampus (CA1). **a** Quantification of PSD95+ puncta on Iba1+ microglial cells, WT-D (46 microglia/5 animals), 3xTg-D (40 microglia/4 animals), WT-S (42 microglia/5 animals), 3xTg-S (43 microglia/5 animals) (2-way ANOVA, Dunnett's post-hoc test: WT-D/3xTg-D, $p = 0.012$; WT-D/3xTg-S, $p = 0.963$). Inset: box plot (minimum, 1st quartile, median, 3rd quartile and maximum), WT-D (8–14 microglia/animal), 3xTg-D (6–10 microglia/animal), WT-S (5–12 microglia/animal), 3xTg-S (7–11 microglia/animal). 2-way ANOVA, Dunnett's post-hoc test. **b** Collapsed z-stack. Scale bar in **b**, 5 μm. **c–i** Single-cell RNA sequencing on WT and 3xTg mice treated with vehicle or SCDi (4 mice per group, pooled). **c** UMAP of all cells. **d** Microglia as percentage of CD45+ tagged cells. **e** Heatmap of WT-D/3xTg-D microglia DEGs ($p$-adjusted $\leq 0.05$). **f** Venn diagram of WT-D/3xTg-D and WT-D/3xTg-S microglial DEG lists. **g, h** Heatmap of Restored (**g**) and New (**h**) genes. **i** Dot plot of average expression ($z$-score) of the top microglia DEGs between 3xTg-D and 3xTg-S: H2-D1, -K1, -T23, -Q6, -M3 genes ($p$-adjusted $\leq 0.05$) and Trem2 ($p$-adjusted $= 0.06$). **j** Venn diagram overlapping the WT-D/3xTg-D microglial DEGs with public datasets from other Alzheimer's disease mouse models. **k–n** qPCR analysis of TNFα (**k**), IL1β (**l**), H2-D1 (**m**), and H2-K1 (**n**) expression in WT ($n = 8$ mice) and SCD1-KO ($n = 3$ mice) microglia treated with Vehicle, LPS, or LPS + SCDi (L + S). TNFα (WT-Vehicle/WT-LPS $p = 0.0001$, SCD1-KO-Vehicle/SCD1-KO-LPS $p = 0.0092$) (**k**), IL1β (WT-Vehicle/WT-LPS $p = 0.0001$, SCD1-KO-Vehicle/SCD1-KO-LPS $p = 0.277$) (**l**), H2-D1 (WT-Vehicle/WT-LPS $p = 0.047$, SCD1-KO-Vehicle/SCD1-KO-LPS $p = 0.901$) (**m**), and H2-K1 (WT-Vehicle/WT-LPS $p = 0.0001$, SCD1-KO-Vehicle/SCD1-KO-LPS $p = 0.059$) (**n**). WT-LPS and SCD1-KO-LPS comparison showed a decrease in TNFα ($p = 0.019$) and a trend for IL1β ($p = 0.075$), 2-way ANOVA, Tukey's post-hoc test. WT-LPS to WT-L + S showed a decrease in TNFα ($p = 0.007$), IL1β ($p = 0.024$) and H2-K1($p = 0.039$) but not H2-D1 ($p = 0.215$) following SCDi treatment. WT-Vehicle to WT-L + S showed an increase in TNFα ($p = 0.0001$) and H2-K1 ($p = 0.038$) but not IL1β ($p = 0.073$) and H2-K1 ($p = 0.626$), 1-way ANOVA, Tukey's post-hoc test. Note no significant differences were found between WT-L + S and SCD1-KO-LPS, 2 tailed $t$-test (comparison with Scd1-KO LPS). $*p \leq 0.05$ $**p \leq 0.01$, $***p \leq 0.001$, $****p \leq 0.0001$. Error bars represent mean ± SEM. SCDi: Stearoyl-CoA desaturase inhibitor. Source data are provided as a Source Data file.

An increasing number of studies show that changes in microglia lipid metabolism mediate their responses in pathological conditions. During aging, for example, microglial subpopulations accumulate lipid droplets that lead to excessive ROS production, release of proinflammatory cytokines and impaired hippocampal synaptic plasticity and spatial memory[49,50]. How SCD-mediated fatty acid desaturation is implicated in regulating microglia activation is just beginning to be understood. Using a mouse model of multiple sclerosis, Bogie and colleagues recently showed that exposing microglia and macrophages to myelin increases their expression of Scd1, leads to their activation and reduces CNS repair. Depleting Scd1 in these cells prevented their activation and promoted myelin regeneration[51]. An interesting parallel exists in the context of AD, where peripheral macrophages treated with Aβ also upregulate Scd1 expression[52]. This is in line with our findings that show that SCDi is able to reverse microglial activation and recover dendritic spine number, dendritic structure, IEG expression and cognition in an amyloidogenic mouse model of AD.

In this study, we found that SCDi greatly dampened microglia activation and had beneficial downstream effects. The single cell resolution of our study allowed us to uncover heterogeneity in Scd gene expression among microglia subtypes and, intriguingly, between microglia from WT and 3xTg mice. Our data shows that Scd plays a role in microglia activation, however, we are just beginning to appreciate the heterogeneity of microglia cells. Using a novel single-cell RNA sequencing strategy, we specifically examined the impact of SCDi on hippocampal microglia. Our analyses uncovered distinct subpopulations of Resting/Homeostatic, Transitional or Activated microglia, with the relative proportions of these subsets being modulated by both the 3xTg phenotype and by SCDi treatment. Two of the Activated microglia subtypes, AR and AIR have been described in multiple recent AD single-cell RNA-sequencing studies[34–36,53]. These previous papers collectively show that both AR and AIR are present in normal young mice, increase with age and even more so in AD. Here, we also identified a third subtype of activated microglia, ASR, that is particularly intriguing because it is the most transcriptomically unique. ASR expressed 311 DEGs compared to the other microglia clusters, and GO analysis revealed that processes related to dendrites, synapses and lipid and cellular metabolism were over-represented in this gene set; this raises the exciting possibility that lipid metabolism may play a role in regulating a type of synapse-associated microglia.

Although ASR did not vary in number between groups, it will be important to develop approaches to study the activity and contribution of this specific microglial subpopulation in health and disease. Interestingly, of the four isoforms of Scd gene found in mice (Scd1–4), Scd1 and Scd2 are the most highly expressed in the brain, and we found that their expression was higher in the Activated than Resting/Homeostatic microglia subtypes. For example, ASR and AR expressed the highest levels of Scd2 and ASR had higher expression of Scd1 compared to the other microglia subtypes. In 3xTg mice, Scd2 was overexpressed in the ASR and AIR subtypes, while Scd1 was overexpressed in ASR. Our findings on the involvement of Scd in microglia polarization support the emerging concept of lipid metabolism being a central regulator of microglial function and are consistent with the idea that microglia undergo metabolic alterations that convert their function from homeostatic to maladaptive in AD and other neurodegenerative diseases.

Intriguingly, the SCDi-mediated recovery of hippocampal dendritic structure, activity-associated IEG expression and learning and memory seen in our study occurred without changes in anxiety, stem/progenitor proliferation, neuronal loss, or amyloid oligomers or tau phosphorylation. Instead, we see main transcriptomic effects on immune genes and synapse genes and dampening of microglia activation. Previous studies have likewise shown that microglia depletion is sufficient to recover synaptic loss and memory in AD mouse models without modifying amyloid burden[54–57]. Together with our data, this suggests that the benefits of microglia depletion are due to removal of active microglia subpopulations specifically. Thus, we hypothesize that there is a tight inter-relationship between the SCDi-mediated effects on microglia and recovery of dendritic structure and learning and memory.

Synapse loss correlates highly with decline in cognitive function during AD[22,23], and microglia are powerful modulators of this process in both health and disease[28,29]. During AD, there is significant and early synaptic loss, likely through dysfunctional synaptic pruning[31,55]. The mechanisms underlying synaptic pruning involve complement-mediated synaptic engulfment by microglia[29,31,58]. Complement proteins and their receptors are upregulated in the context of AD[31,59,60], and their inhibition is sufficient to rescue cognitive impairment, synaptic and neuronal loss[31,58]. Our data shows that MHC-I genes were among the most highly up-regulated genes between 3xTg-D and WT-D, most strongly down-regulated genes by SCDi, and were expressed

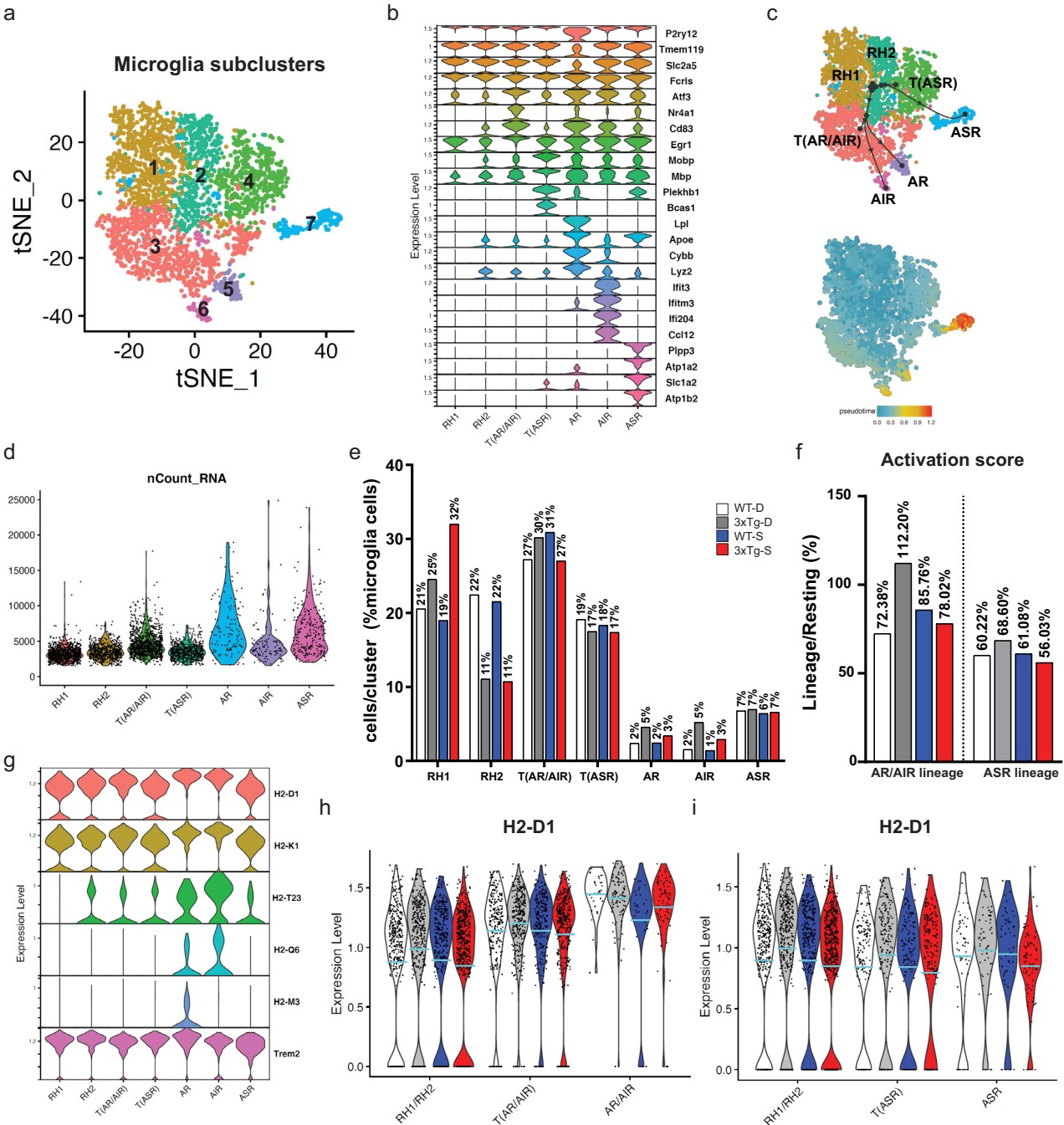

**Fig. 6 Single-cell analysis uncovers SCDi-mediated changes to the cellular landscape of microglia. a** tSNE plot showing the microglia population can be subclustered into seven microglial subtypes/activation states (subclusters 1–7). **b** Violin plots of top 4 DEGs used to define the microglia cluster (P2ry12, Tmem119, Slc2a5, Fcris) followed by the top 4 subcluster-specific DEGs. Subclusters were annotated on the basis of prior single-cell RNA sequencing studies and additional strategies as described in the sections "Results" and "Methods": Resting/Homeostatic (RH1, RH2, subclusters 1,2), Transitioning (T(AR/AIR), subcluster 3), Transitioning (T(ASR), subcluster 4), Active Response (AR, subcluster 5), Active Interferon Response (AIR, subcluster 6), Active Synapse-associated Response (ASR, subcluster 7). **c** PAGA-tree trajectory inference and pseudotime analysis of the microglia cluster, showing microglia subclusters annotated as "Active" (AR, AIR, ASR) have increased pseudotime transition. **d** Violin plot of nCounts across all microglia subclusters, showing the microglia subclusters annotated as "Active" (AR, AIR, ASR) have higher numbers of unique molecular identifiers (UMI) than "Resting" microglia (RH1, RH2). **e** Percentages of total microglia found in each subcluster in each experimental group. **f** Activation scores for the AR/AIR and ASR lineages for each experimental group. Activation scores were calculated as the lineage-positive cells ("Transitional" + "Active") expressed as percentage of the resting cells (RH1 + RH2). 100% indicates an equal proportion of lineage-positive to resting cells. **g** Violin plots of showing relative expression of the top 6 SCDi-regulated genes (between 3xTg-D and 3xTg-S) across microglial subclusters. **h, i** Violin plots of expression levels of H2-D1 in the AR/AIR lineage (**h**) and ASR lineage (**i**) in each experimental group. Blue lines identify the average expression level. Legend of experimental groups in (**e**) applies to panels **e**, **f**, **h**, **i**. See Supplemental Figs. 6–8 and Supplemental Data Files 2, 3.

at highest levels by the activated microglia subclusters. MHC-I genes are involved in antigen presentation and activation of the adaptive immune response but also may play a role in synaptic pruning[38] and learning and memory[39]. MHC-I molecules colocalize with the complement component C1q, suggesting a cooperative role for these genes in synaptic regulation[61]. In AD, C1q depletion is associated with reduced microglia/macrophage activation and AD mice lacking C1q display reduced synapse loss, supporting a role for C1q in mediating synapse removal[58]. Moreover, C3 depletion in a mouse model of AD significantly reduces synapse loss and promotes cognition, despite accumulation of the amyloid burden[62]. Of particular relevance here, Dagher et al. showed that a ~30% reduction in microglia number in 3xTg mice improved cognition without significant changes in amyloid burden, suggesting that microglia might contribute to cognitive dysfunction via Aβ-independent mechanisms[63].

Together our data suggest a Scd-MHC-I pathway mediates dendritic structure and learning and memory deficits in AD. Directly testing this hypothesis will require the development of more targeted tools that can inhibit Scd-expressing activated microglia cell subpopulations selectively.

Microglia are taking their place as key regulators of the pathogenesis of AD and other neurodegenerative diseases. As such, they are an attractive target for new disease-modulating drugs but also as potential biomarkers of clinical progression and drug-engagement. Using single-cell RNA sequencing, we and others have identified microglia subtype signatures that can hopefully be used to delineate biological states in the study of microglia dynamics. Identification of robust markers for different microglia states will allow for the development of tools to visualize microglial activation, inflammation, and disease progression. More broadly, our study places fatty acid metabolism and specifically SCD as a target to modulate core AD disease hallmarks. Encouragingly, inhibitors of human SCD have already been approved for clinical trials in obesity (study identifier; NCT02647970) and more recently in Parkinson's disease (Yumanity); our data support their testing in AD patients as well.

## Methods

**Mice.** Experiments were approved by the *Institutional Animal Care Committee* of the Centre de Recherche du Centre Hospitalier de l'Université de Montréal (CRCHUM) or Université de Quebec à Montreal (UQAM) following the Canadian Council of Animal Care guidelines.

**3xTg-AD mice and strain controls.** 3xTg mice (Jackson Laboratory MMRRC stock #: 034830) and their WT strain controls (B6129SF2/J, Jackson Laboratory stock #: 101045) were purchased from Jax mice. Briefly, 3xTg mice were originally derived by Oddo and colleagues by co-microinjecting two independent transgenes encoding human APP_Swe and the human tau_P301L (both under control of the neuron-specific mouse Thy1.2 regulatory element) into single-cell embryos harvested from homozygous mutant PS1_M146V knock in (PS1-KI) mice[14]. Wildtype mice are the PS1-knock-in background strain (C57BL/6J × 129S1[14]). All mice were bred in-house, maintained in identical housing conditions (22 °C, 50% humidity, 12 h Light/Dark cycle OFF 10 a.m. ON 10 p.m.) and given free access to a standard rodent diet (#2018, Harlan Teklad) and water. 3xTg mice undergo a progressive increase in amyloid beta peptide deposition, with intracellular immunoreactivity being detected in some brain regions as early as 3–4 months. Synaptic transmission and long-term potentiation are demonstrably impaired in mice 6 months of age. Evidence of gliosis and inflammation is present by at least 7 months[64]. Between 12 and 15 months aggregates of conformationally altered and hyperphosphorylated tau are detected in the hippocampus[14,15]. Female mice were used for all experiments and sex and age matched.

**SCD1-KO mice.** Scd1 knockout (Scd1-KO) mice were originally generated by the Ntambi laboratory (University of Wisconsin-Madison, Madison, USA)[65]. These mice were generated using a targeting vector to replace all six exons of the endogenous Scd1 gene with a neomycin-resistant cassette. This construct was electroporated into 129/Sv-derived embryonic stem (ES) cells. Heterozygous (Scd1-Het) mice were bred and housed (26 °C, 50% humidity, 12 h Light/Dark cycle ON 7 a.m. OFF 7 p.m.) in the Université de Quebec à Montreal (UQAM) animal facility to produce the WT and KO genotypes used in this study. Genotypes were

confirmed by PCR amplification of mouse tail genomic DNA using the following primers: primer oIMR5361 (5′-GGGTGAGCATGGTGCTCAGTCCCT-3′) located in exon 6, primer oIMR6982 (5′-ATAGCAGGCATGCTGGGGAT-3′) located in the neo gene (425-bp product, targeted allele) and primer oIMR5362 (5′-CACA CCATATCTGTCCCCGACAAATGTC-3′) located downstream of the targeting gene (600-bp product, wild-type allele). PCR conditions were 28 cycles, each of 45 s at 94 °C, 30 s at 62 °C and 45 s at 72 °C.

**SCD inhibitors.** Two SCD inhibitors were used in this study: ab142089 (Abcam, C20H22ClN3O3, 4-(2-Chlorophenoxy)-N-[3 [(methylamino)carbonyl]phenyl]-1-piperidinecarboxamide, M.W. 387.87) and CAY10566 (Cayman Chemical, C18H17ClFN5O2, 3-[4-(2-chloro-5-fluorophenoxy)-1-piperidinyl]-6-(5-methyl-1,3,4-oxadiazol-2-yl)-pyridazine, M.W. 389.8).

**In vivo surgical procedures**

*Intracerebroventricular (ICV) osmotic pumps.* For ICV infusions of SCDi or vehicle, mice were locally injected with buprivacaine (Hospira) and operated under iso-flurane anesthesia (Baxter). Brain cannulae attached to Alzet osmotic pumps were stereotaxically implanted at 0.0 mm antero-posterior and 0.9 mm lateral to Bregma and the pumps placed under the back skin according to manufacturer's instructions. The 28-day Alzet osmotic pumps used in these experiments (0.11 μl/h infusion rate, model 1004; Durect) were primed for 48 h and begin pumping immediately when implanted. Abcam or Cayman SCD inhibitors were dissolved in DMSO (Sigma-Aldrich) and infused at a final concentration of 80 μM in sterile aCSF (148 mM NaCl, 3 mM KCl, 1.7 mM MgCl₂, 1.4 mM CaCl₂, 1.5 mM Na₂PO₄, 0.1 mM NaH₂PO₄). Vehicle pumps contained the same volumes as the SCDi pumps (0.8% DMSO/aCSF).

**Fatty acid extraction and analysis by gas chromatography–flame ion detection (GC–FID).** Total lipids were extracted from periventricular tissue samples using the Folch method[66]. Sample weights (2.31–5.31 mg) were used to calculate the amount of standard (30–70 μg) of 1-O-Heptadecyl-2-Acetyl-sn-Glycero-3-Phosphocholine (17:0 PC; Sigma-Aldrich cat# 850360P) previously diluted in 10 mL of chloroform was added as an internal standard prior to extraction. Saponification of the fatty acids was performed using KOH-methanol, protonation followed with 12 N HCl and methylation with 12% BF₃-methanol (Thermo Fisher Scientific cat# AC402765000), as described previously in Plourde et al.[67,68].

The fatty acid composition of the periventricular tissue samples was analyzed by gas chromatography coupled with a flame ionization detector (model 6890; Agilent). 1 μl of the sample was injected in splitless mode at 250 °C. The methylated fatty acids were carried by helium at a pressure of 107 kPA through a 50 m BPX-70 fused capillary column (0.22-mm diameter, 0.25-μm film thickness; SGE). A thermal gradient was applied on the column as described in Chevalier et al.[69]. Fatty acids were then detected at the end of the column by the flame ionization detector at 260 °C. The total run time was 61.75 min per sample. Nu-Check-Prep standards (GLC-569) were used to identify 45 different fatty acid methyl esters and the chromatogram analysis was performed using OpenLab CDS ChemStation version C01.10.

**Whole hippocampus bulk RNA sequencing**

*RNA extraction, library preparation and sequencing.* Mice were euthanized using a lethal dose of ketamine (Bimeda-MTC)/xylazine (Bayer Healthcare). Hippocampi from unoperated or pump-implanted mice were dissected and immediately flash frozen on dry ice. Total RNA was extracted with TRIzol (Invitrogen) and chloroform (Invitrogen). RNA was then purified with the RNeasy mini kit (Qiagen) according to manufacturer's instructions. 400 ng of total RNA was used for library preparation. RNA quality control was assessed with the Bioanalyzer RNA 6000 Nano assay on the 2100 Bioanalyzer system (Agilent technologies) and all samples had a RIN above 8. Library preparation was done with the KAPA mRNAseq Hyperprep kit (KAPA, Cat no. KK8581). Ligation was made with 38 nM final concentration of Illumina dual-index UMI (IDT) and 11 PCR cycles was required to amplify cDNA libraries. Libraries were quantified by QuBit and BioAnalyzer DNA1000. All libraries were diluted to 10 nM and normalized by qPCR using the KAPA library quantification kit (KAPA; Cat no. KK4973). Libraries were pooled to equimolar concentration. Sequencing was performed with the Illumina Nextseq500 using the Nextseq High Output 75 cycles kit using 2.6 pM of the pooled libraries. 20–30M single-end PF reads were generated per sample. Library preparation and sequencing was made at the Institute for Research in Immunology and Cancer's Genomics Platform (IRIC).

*Bio-informatics.* Sequences were trimmed for sequencing adapters and low quality 3′ bases using Trimmomatic version 0.35 and aligned to the reference mouse genome version GRCm38 (gene annotation from Gencode version M23, based on Ensembl 98) using STAR v2.5.1b. Group data was analyzed pairwise and normalized using the DEGES/DESeq2 generalized linear model. Differentially expressed genes were identified with the TCC implementation of likelihood ratio test within DESeq2 with a *p*-value < 0.01 threshold. Pheatmap v1.0.12 was used to produce heatmaps.

GO enrichment analysis on DEGs was performed using ShinyGO v0.741[70], significance was set to FDR ≤ 0.05 and limited to the top 20 most significant categories for all analyses. The network plot (Fig. 1f) shows the relationship between enriched

pathways. Two pathways (nodes) are connected if they share 20% or more genes. Darker nodes are more significantly enriched gene sets, bigger nodes represent larger gene sets, thicker edges represent more overlapped genes. For the hierarchical clustering tree (Supplemental Figs. 6g and 7e–g) summarizing the correlation among significant pathways, the pathways with many shared genes are clustered together. Bigger dots indicate more significant *p*-values. For identification of the DEGs related to "Lipid", "Immune response", or "Synapse", gene set analysis was first performed on DEGs using GSEA 4.0.3[71], and enriched gene sets relating to Lipids (lipid, fat, sterol), Immune responses (inflam, cytokine, immun, neutro, antigen, leukocyte, chemotaxis, lymphocyte, interleukin, myeloid, interferon, chemokine), and Synapses (synap, dendri, neurotrans, spine) were curated using the keywords in parentheses.

### Hippocampus single-cell RNA sequencing
*Dissociation protocol.* Mice received a lethal dose of ketamine (Bimeda-MTC)/ xylazine (Bayer Healthcare) and were then perfused transcardially with phosphate-buffered saline (PBS) followed by rapid dissection of the hippocampus from the pump-implanted side into ice cold HBSS. Pools of 2 hippocampi were transferred to 1 ml of a warm (37 °C) Papain/DNAse I solution and mechanically dissociated 20 times with a plastic pipette before being incubated for 10 min at 37 °C. The same step was repeated with a P1000, followed by another dissociation with the P1000 and the addition of Ovomucoid (1:1 in volume) to stop the enzyme activity. Cells were triturated again 20 times with a P1000 and then a P200 and Base media (BM) (3:1 DMEM /F12 + GlutaMAX) was added to dilute the cells to 10 ml. Cells were through a 40 μm cell strainer then centrifuged at 500×*g* for 10 min at 4 °C without brakes. The supernatant was removed (leaving 1 ml) and a red blood cells lysis buffer diluted 1:1 with BM was added up to 6 ml and incubated for 1 min. After incubation, BM was added up to 14 ml to wash the lysis buffer, and cells were centrifuged as above. After centrifugation, the supernatant was removed, and pellet resuspend in BM up to 6 ml. The cell suspension was again passed through a 40 μm cell strainer, followed by centrifugation as before. As much as possible of the supernatant was removed and the remainder incubated with 100 μl of Dead Cell Removal Beads (Miltenyi Biotec) for 15 min at RT. Live cells were collected using MS selection columns (Miltenyi Biotec) according to the manufacturer's instructions.

*Multiplexing and sequencing.* Cells were centrifuged, the supernatant was removed, and the cells resuspended in 400 μl of FBS (staining buffer, BD) before incubation for 5 min at 37 °C in 1:200 dilutions of Draq7 (300 μM) and Calcein AM (2 mM) to stain live and dead cells. Cells from each sample were pooled to obtain 4 mice per group, then the cells were labeled with CD45 mouse immune multiplexing kit (BD Biosciences, USA), loaded on a hemocytometer (C-chip, DHC-N01-5, (INCYTO) and the concentration and viability was measured in the BD Rhapsody Single-cell Analysis system (BD Biosciences). Then, 10,000 cells per sample were immediately loaded into the cartridge ready for single-cell capture. The viability of cells in all samples was 80–95%. Single-cell capture was performed on the BD Rhapsody (BD Biosciences), the DNA from each cell attached to a barcoded bead was subsequently retrieved and libraries were prepared using the whole transcriptome analysis (WTA) amplification kit (BD Biosciences) according to the manufacturer's recommendations[72]. The quality of WTA and Sample tag libraries were examined with a DNA screentape D5000 HS on a TapeStation 2200 (Agilent Technologies, Santa Clara, CA, USA) and the concentration was measured on a QuBit 3.0 fluorometer (Thermo Fisher Scientific, Canada). Subsequently, WTA and Sample Tag libraries were pooled in a ratio of 98.8%:1.2%, initially sequenced on an iSeq 100 (Illumina Inc., San Diego, CA, USA) to confirm quality of libraries, and then sequenced for paired-end 100 bp sequencing on a NovaSeq 6000 (Novaseq 6000 S4 reagent kit (200 cycles), Illumina Inc., San Diego, CA, USA) at the Next-Generation sequencing Platform, Genomics Center, CHU de Québec-Université Laval Research Center, Québec City, Canada. The coverage was ~1300M paired-end reads.

*Bio-informatics.* Sequencing fastq files were processed with the BD Rhapsody WTA Analysis Pipeline on the SevenBridges application (BD Biosciences) using the mm10 reference genome with gencodevM19-20181206 transcriptome annotation. The unfiltered expression matrix and sample tag call matrix for multiplexing were recovered and processed using standard scanpy v.1.4.3 analysis pipeline[73] SCANPY: large-scale single-cell gene expression data analysis in the Genetics & Genomics Analysis Platform on Compute Canada's Arbutus cluster (https://genap.ca/). In total, 24,154 genes across 20,959 cells were analyzed, with 31,681 mean reads/cell, 5336 mean UMI/cell, and 1866–2502 cells/tag for the 4 multiplex tags. h5Anndata object was generated after data normalization and scaling, dimensionality reduction, cluster and marker identification and imported into Seurat 4.1.0[74] for further analysis. Cells were filtered for UMI count (min 1000) and mitochondrial content (max 27%), resulting in a final cell count of 17,983. For the total microglia cluster (clusters 1, 8, 20), the average read count was 4724 (Supplemental Fig. 6c) and the average mitochondrial content was 6.1% (Supplemental Fig. 6d).

Automatic cell type annotation was conducted using scMCA mouse cell atlas R package v0.2.0.[75] and the "Single-cell transcriptomics of 20 mouse organs creates a Tabula Muris" trained model in scibet R package v0.1.0[76] and manually annotated by confirming results from automatic annotation and markers identified with the

*FindMarkers* function when consensus was not evident between automatic methods. Microglial cells were subsetted and reanalyzed to identify subpopulations on the 2000 variable features identified by *FindVariableFeatures*. Data scaling and PCA analysis was conducted and the top 10 PCA dimensions, as determined with elbow plot, were used to *FindNeighbors* and *RunTSNE*. FindClusters was run with a sensitivity of 0.4 to identify subpopulations within the microglia subset. *FindMarkers* was used to identify the transcriptomic signature of each subtype and markers were compared with previously published data[34–36,53].

Trajectory inference was conducted using the dynoverse R packages dyno 0.1.2: Set of packages for doing trajectory inference (TI) on single-cell data. https://github.com/dynverse/dyno[77]. Briefly, count and expression matrices were extracted from the Seurat object and wrapped into a dynwrap object using *dynwrap::wrap_expression.* grouping and dimensionality reduction information, these were then added to the dynwrap object using *add_grouping* and *add_dimred* respectively. TI methods were selected using the *guidelines_shiny* tool for our specific dataset. TI methods retained for analysis were slingshot[78], PAGA-tree[41], multiple spanning tree (MST), and wishbone[79]. While every method provided specific insight into the data structure upon TI analysis using *dynwrap::infer_trajectory*, PAGA-tree best recapitulated the structure of other methods combined in our data. Pseudotime visualization was thus obtained using *calculate_pseudotime* on the PAGA-tree model object.

Gene expression mapping was done using *Nebulosa::plot_density*[80] *Nebulosa: Single-Cell Data Visualization Using Kernel Gene-Weighted Density Estimation.* R package version 1.0.0, https://github.com/powellgenomicslab/Nebulosa). Differential gene expression between treatment groups was done with Seurat::FindMarkers and specifying treatment pairs to compare. Violin plots were generated in Seurat and represent the natural log of the number of transcripts after normalization.

### Comparison with publicly available mouse microglia datasets
*Bulk microglia data sets.* The protein coding transcripts of the combined matrix on GEO from Xia et al. (GSE158153) was extracted and all replicated were analyzed using the same methodology as for our data set using the TCC R package. All App-SAA samples were analyzed as replicates, irrelevant of their methoxy-X04 status (*n* = 20 total) and compared to control wildtype samples (*n* = 10) using tcc's implementation of Deseq2 for both normalization and DEG testing with three iterations with a *p*-value ≤ 0.01 threshold. Similarly, raw fastq files for WT-PBS and 5XFAD-PBS from Baik et al. (PRJEB32503) were processed using our standard RNAseq pipeline[40]. Briefly, the RNA-sequencing data was run in a bioinformatics pipeline using the Compute Canada clusters (assigned to Dr. Tétreault). Alignment was performed using HiSAT2 v2.1.0[81] against reference genome mm19. Read counts were obtained with htseq-count[82] and differential expression analysis were performed with the Bioconductor R package TCC v1.24.0[83]. Group data was analyzed pairwise and normalized using the DEGES/DESeq2 generalized linear model. Differentially expressed genes were identified with the TCC implementation of likelihood ratio test within DESeq2 with a *p*-value < 0.01 threshold.

*Single-cell microglia datasets.* The Seurat-normalized filtered counts matrix and metadata from Sierksma et al. [36] (GSE142267) was imported into Seurat. DEGs were obtained by running the FindMarkers function between specified groups (APP_WT vs. APP_TG) at 11 months. Similarly, for Sala Frigerio et al. the Seurat-normalized filtered counts matrices and metadata from Sala Frigerio et al.[34] (GSE127893) for the two available datasets were imported into Seurat for reanalysis. Only WT vs. APP_NL-F-G 12 months old animals from the KW dataset were compared using the FindMarkers function to identify DEGs between microglial cells in hippocampus populations. DEG significance cut-off was set to *p*-adj ≤ 0.05 for single-cell RNA sequencing data.

DEG lists from all data sets analyzed were compared using molbiotools (molbiotools.com/listcompare) to determine the common genes.

### Microglia cultures
Primary microglia were harvested from whole brains of postnatal day 1 Scd1-WT or Scd1-KO pups. Following the dissection of the brain, meninges were carefully removed in an ice-cold solution of HBSS 1X without Ca²⁺ and Mg²⁺ (Wisent, cat# 311- 512-CL). The brain was chopped using a scalpel, passed through a 25 and 22 G syringe needle twice, then filtered through 100 μm cell strainer in Dulbecco's modified Eagle medium (DMEM), high glucose without sodium pyruvate (Invitrogen cat# 11965118) containing 10% heat-deactivated FBS (Wisent cat# 085150) and 1% penicillin and streptomycin (P/S) (Wisent cat# 450-201-EL). Cells from each brain were resuspended in media (DMEM + 10% FBS + 1%P/S) and cultured in a T-75 flask in a 37 °C incubator with 5% CO₂. After 5 days in culture, the media was changed to remove debris. The media was subsequently changed 2 times a week until microglia isolation. Once a layer of astrocytes reached 95–100% confluence (11–13DIV) the microglia grow on top, microglia began detaching after an additional 3DIV and were isolated by incubating flasks on a rocker for 1 h at 37 °C with 5% CO₂. Microglia were then plated on a 48-well plate coated with poly-D-lysine (Sigma Aldrich, cat# P9404) at a cell density of 100,000 cells/well and incubated for 48 h before the experiment to allow microglia to enter a "homeostatic/resting state". This protocol yields approximately ~98% microglia purity in our lab. Primary microglia were treated for 6 h with 25 ng/mL of LPS (Sigma Aldrich, cat# L4516) diluted in PBS 1× (Wisent, cat#311-

010-CL), or 5 µM of SCD1 inhibitor (Abcam cat# ab142089) diluted in DMSO. Treatments were stopped by adding TRIzol (Invitrogen cat# 15596018) to the cells followed by RNA extraction with chloroform (Invitrogen cat# 15593031).

**Real-time quantitative PCR.** RNA was quantified using a NanoDrop™ 2000/2000c Spectrophotometer (Thermo Fisher Scientific). cDNA was prepared from 1 µg of total RNA using a mix of 1× reverse transcriptase buffer (from 10× stock: 1 M Tris pH 8.3, 1 M MgCl₂), 25 mM dNTP mix (Invitrogen cat# 10297-018), 0.5 M dithiothreitol (Invitrogen cat# R0861), 20U RNAse inhibitor (Invitrogen cat# N8080119) and 100U M-MLV Reverse Transcriptase (Invitrogen cat# 28025-013). Samples were incubated at 37 °C for 1 h. cDNA was diluted to a final concentration of 4 ng/µL in RNAse–DNase-free water (Wisent, cat# 809-115-CL). Real-time quantitative PCR was performed using SensiFAST 2X SYBR Lo-ROX kit Master mix (Bioline cat#BIO-94050) on the QuantStudio3 (Applied Biosystems). Primers (Supplemental Table 1) were used at 16 µM. Gene expression of target mRNAs was calculated using the Comparative Ct method and was normalized to expression levels of the housekeeping gene 18S.

**Golgi-Cox staining, imaging and quantification.** Mice were deeply anesthetized with isoflurane and immediately decapitated, the brain was removed, and the contralateral half was cut at 5 mm from the olfactory bulbs and 4 mm from the brain stem in a mouse brain mold to expose the hippocampus. Golgi staining was performed with the slice Golgi kit (Bioenno Tech, USA). Slices between Bregma −1.06 to −2.20 mm were post-fixed in fixative solution for 1 h at 4 °C and then in fresh fixative solution for an additional 48 h at 4 °C before slicing on a vibratome at 150 µm. Briefly, sections were washed three times with PB pH 7.4. Free-floating sections were then stained in the impregnation solution for 24 h and then in fresh impregnation solution for another 4 days at room temperature on a rocker. Then sections were washed once with PBS-T to remove impregnation solution. Subsequent washes were performed with PBS 1× pH 7.4. Golgi staining was performed by incubating sections in solution C for 1 min and then in post-staining solution D for 2 min. Sections were dehydrated in a series of ethanol baths (50%, 80%, 95% and 100%) followed by two Xylene washes. Sections were then mounted with Permount (Thermo Fisher Scientific, Canada) and allowed to dry overnight before imaging.

Dendritic spine density was assessed by counting spines found on secondary and tertiary dendrites of neurons in the dentate gyrus and the CA1 of the hippocampus on a light microscope at ×100 equipped with a color camera (Olympus BX43). To assess dendritic complexity, Z stacks were taken at 0.2 µm steps at ×62 in brightfield (Olympus X81). Stacks were processed in Fiji v1.52p[84]. Using the Segmentation (Simple Neurite Tracer) plug-in, dendritic trees were traced to obtain total dendritic length and a Sholl analysis was performed using 10 µm concentric rings using the center of the soma as the reference point. Quantifications in DG or CA1 were performed on the contralateral (side-opposite the ICV pump), approximately two neurons from (4 sections per animal n = 4 mice per treatment group) for a total of ~24 neurons per group. All quantifications were done by a blinded experimenter for further details refer to figure legends.

**Cresyl Violet staining, imaging and quantification.** Four 40 µm vibratome sections from n = 5 animals per group (Bregma −1.06 to −2.20 mm) of the hippocampus were dried on slides, fixed with PFA 4% for 10 min and stained with cresyl violet acetate (0.1%, Sigma-Aldrich) for 15 min. Slides were quickly destained in a series of ethanol baths (50%, 70%, 95%, 100%, 15 s per bath), cleared with Citrisolv (Thermo Fisher Scientific, Canada) for 10 s and coverslipped with Permount (Thermo Fisher Scientific, Canada).

Quantification of the neuron density was performed live on a light microscope equipped with a color camera at ×40 (Olympus BX43) from one field of the dentate gyrus or CA1 per section. For the measurement of granule and pyramidal cell layers, the dentate gyrus and the CA1 of the hippocampus were imaged on a light microscope equipped with a color camera at ×40 (Olympus BX43) and the thickness was measured using the measure tool in Fiji v1.52p[84]. All quantifications were done by a blinded experimenter.

**Immunohistochemistry, imaging and quantification.** Immunohistochemical procedures were performed as detailed previously[85,86] for mouse anti-human Ki67 (1:100, BD Biosciences), rabbit anti-GFAP (Dako Diagnostic, 1:1000), and guinea-pig anti-DCX (Millipore, 1:3000). For rabbit anti-Iba-1 (Wako, 1:400) and mouse anti-post synaptic density-95 (Millipore, 1:160) the sections were blocked overnight in 10% normal donkey serum/0.2% triton x at 4 °C followed by overnight incubation in the primary antibodies at 4 °C. Fluorescence primary antibodies were detected using species-appropriate Alexa Fluor conjugate secondary antibodies, Alexa 488 Donkey anti-rabbit (H + L) (A21206, Invitrogen, 1:1000), Alexa 555 Goat anti-guinea pig (H + L) (A21435, Invitrogen, 1:1000), Alexa 647 Donkey anti-mouse (H + L) (A31570, Invitrogen, 1:1000). For brightfield immunohistochemistry, rabbit anti-Abeta (37–42, NEB, 1:100) and mouse anti-Abeta 1–16 (Covance, 6E10, 1:200) primary antibodies were detected using species-appropriate biotinylated secondary antibodies biotinylated goat anti-mouse (H + L) (115-065-146, Jackson Immuno Research, 1:1000) or biotinylated goat anti-rabbit (H + L) (111-065-144, Jackson Immuno Research, 1:1000), and the signal was amplified using

the avidin–biotin–peroxidase system (VectaStain ABC Kit, Vector Laboratories) and revealed using a 3,3-diaminobenzidine (DAB)-containing solution (Sigma-Aldrich).

Ki67, GFAP, and DCX was imaged on a motorized Olympus IX81 microscope. For quantification of neurogenesis markers expressed by SVZ and dentate gyrus cell populations, 4–6 sections at 120 µm intervals through the striatal SVZ or dentate gyrus were quantified. Counts were limited to the subventricular zone and subgranular zone respectively. All Abeta and Tau sections were scanned with a 20 × 0.75 NA objective with a resolution of 0.3225 µm (BX61VS, Olympus, Toronto, Ontario). Quantification of amyloid beta Abeta (37–42) and Abeta (1–16) was performed in Fiji v1.52p[84] using the process and analyze particle functions to obtain arbitrary units (arb. units) per animal. For colocalization of PSD95 and Iba-1, sections were imaged on a confocal Leica SP5 using ×63 oil 1.4 NA objective with 1× zoom and 2048 × 2048 resolution, 400 Hz. 20–30 µm z-stacks were acquired at 0.4 µm z-step size, using sequential scans with 2× frame averaging and 2× line averaging for both the 488 nm (Iba-1) and 647 nm (PSD95) wavelengths. For quantitation, image brightness was adjusted systematically to all stacks and 3D rendering of confocal images was performed using 3D project in Fiji (v1.52p). Only puncta larger than approximately 0.4 µm were counted using the manual cell counter plug-in. Iba-1+ microglial cells in CA1 with complete arborization were quantified, WT-D (n = 46 from 5 animals), 3xTg-D (n = 40 cells from 4 animals), WT-S (n = 42 cells from 5 animals), 3xTg-S (n = 43 cells from 5 animals). The experimenter was blinded to the genotype and treatment during image acquisition, processing and quantification of all IHC experiments.

**Western blotting.** Western blotting was performed as previously described[86]. The following antibodies were used: mouse PHF1 (p-Tau(S396/404))(Peter Davis, 1:100), mouse AT8 (p-Tau(S202/T205))(Thermo, 1:100), mouse AT180 (p-Tau(T231)) (Thermo, 1:100), mouse anti-Abeta 1–16 (Covance, 6E10, 1:1000), rabbit anti-Abeta (37–42, NEB, 1:500) in conjunction with HRP-conjugated secondary HRP Goat anti-rabbit (111-035-144, Jackson Immuno Research, 1:5000), HRP Goat anti-mouse (Bio-Rad, 170-6516, 1:5000). Secondary antibodies were detected using Clarity (Bio-Rad) and ChemiDoc (Bio-Rad). Quantitative densitometry of bands was performed using Image Lab version 4.1 (Bio-Rad).

**Behavioral testing**

*Open field test.* As As of anxiety-like behavior, mice were tested in the open field test. The open field consists of a Plexiglass box (50 × 50 × 30 cm) in a lit room. Each mouse was placed in the middle of the novel arena and allowed to explore the field for 5 min. Movement in the field was recorded and tracked by an overhead video camera connected to a PC with Ethovision XT software (Med Associates, Inc., St Albans, VT, USA).

*Elevated plus maze.* The apparatus consists of two closed arms that oppose two open arms in a plus design (Med Associates, Inc., St Albans, VT, USA). Decreased time spent in the open, exposed arm is an indicator of increased anxiety-like behavior. The apparatus is placed 60 cm above the floor and has a video camera fixed overhead. Each mouse was placed in the middle of the maze facing the open arm opposing the experimenter. Movement in the maze was recorded and tracked for 5 min by an overhead video camera connected to a PC with Ethovision XT software (Med Associates, Inc., St Albans, VT, USA).

*Light dark box.* The light dark box consisted of a shuttle box half lit and half dark. Decreased time spent on the lit side is an indicator of increased anxiety-like behavior. Each mouse was placed in the dark side of the box and movement was recorded and tracked for 5 min by an overhead video camera connected to a PC with Ethovision XT software (Med Associates, Inc., St Albans, VT, USA). The number of 4 paw exits was manually counted using the video by a blinded experimenter.

*Morris water maze.* The water maze consists of a pool (100 cm in diameter) surrounded by extra maze cues delimiting four quadrants (NE, NW, SE, SW). The water temperature was maintained at 25 ± 1 °C (24.5 °C = 76 °F). Mice were trained for 6 trials per day over 4 days. On Day 5, the platform was removed for a probe trial. Starting positions were chosen using a pseudorandom algorithm and changed for each trial. The maximum allotted time for each trial was 60 s and mice were considered by the software to have arrived on the platform if they were in the platform zone for 2 s. Mice were allowed to remain on the platform for 20 s; mice that did not find the platform were guided there and allowed to remain for 20 s. Following each trial, mice were lightly dried and placed on a heating pad until their next trial. The inter-trial interval ranged from 15 to 20 min.

**Statistical analyses.** All statistical analyses were performed using GraphPad Prism, Version 8 (GraphPad Software, Inc). A 2 × 2 experimental design (strain and treatment) was used for most experiments, and therefore analyzed by 2-way ANOVA followed by either Dunnett's (comparisons versus a single control group) or Tukey's (all group comparisons) multiple comparison post-hoc tests, as indicated in the legends. For the Morris Water Maze (Fig. 3a), a 2-way ANOVA with repeated measures was used for analyzing thee escape latency learning curves.

For the microglia culture experiment (Fig. 5k–n), the WT and Scd1-KO strains were treated with 3 treatments (WT: Vehicle, LPS, LPS + SCDi) versus 2 treatments (Scd-KO: Vehicle, LPS), precluding use of a single test. A primary 2-way ANOVA (Tukey's post-hoc test) was therefore used to compare the Vehicle and LPS treatments on WT and Scd1-KO mice, and secondary tests were used for the additional comparisons involving the WT (LPS + SCDi) group: the WT (LPS + SCDi) group was compared with the two other WT groups by 1-way ANOVA (Tukey's post-hoc test), and with the Scd1-KO (LPS) group by $t$-test. Error bars represent mean ± standard error of the mean (SEM). Significance level was set at $p \leq 0.05$ or $p \leq 0.01$ as indicated in figure legends.

**Reporting summary**. Further information on research design is available in the Nature Research Reporting Summary linked to this article.

## Data availability

The datasets generated during this study are available at GEO, accession number GSE167605. All the other data are available within the article and its Supplementary Information. Source data are provided with this paper.

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

## Acknowledgements

We thank the Genomics Platform of the Institute for Research in Immunology and Cancer (IRIC) for the bulk hippocampal sequencing, the Next-Generation Sequencing Platform of the Genomics Center at CHU de Québec-Université Laval Research Center for the single-cell sequencing, the molecular pathology core facility of the CRCHUM for slide scanning, and Compute Canada (www.computecanada.ca), GenAP (genap.ca) and its private Galaxy instance (galaxy.org) for bioinformatic resources. This work was funded by operating grants to K.J.L.F. from the Canadian Institutes of Health Research (CIHR), the Alzheimer Society of Canada, and the Natural Sciences and Engineering Research Council (NSERC). L.K.H was supported by a Spark Award fellowship from the Canadian Alzheimer Society Research Program (ASRP). M.T. holds a FRQS Junior 1 salary award and funding from the Fondation Courtois.

## Author contributions

L.K.H. developed the concept, carried out experiments, analyzed data, and co-wrote the manuscript. G.M.B. and M.T. performed bioinformatic analyses on the whole hippocampus and single-cell RNA sequencing data sets. C.L.M contributed to the whole hippocampal RNA sequencing and Golgi staining and analysis. F.P helped develop the single-cell dissociation and RNA sequencing protocol. M.A. performed GC-FID in collaboration with A.V. and M.P. A.L. and C.M. contributed the Scd1-KO mice. S.E.J. performed ICV surgeries. A.A. contributed to the Golgi experiments and performed amyloid and tau immunostaining and Western blots. K.J.L.F. developed the concept, performed ICV surgeries, analyzed data, and co-wrote the manuscript. All authors revised the manuscript.

## Competing interests

The authors declare no competing interests.
