## [Peer Review File · Nature Communications]

Reviewers' Comments:

Reviewer #1:

Remarks to the Author:

This work by Hamilton et al., is a compelling, multifaceted study on an important topic, connecting fatty acid metabolism and neurodegeneration in Alzheimers disease. The research presented in this manuscript builds significantly on the groups 2015 work (PMID: 26321199) analyzing transcriptomic differences between WT and symptomatic 3x-Tg-AD mice. It connects the role of SCD and its inhibition in the downstream cellular responses of microglia and neurons to amyloid+tau deposition in a credible model of AD. There are numerous strengths, including the detailed TP and phenotyping, the behavioral readouts, the dendritic quantification and especially the detailed and sometimes novel methods for classifying subpopulations of microglia and their differential TP changes in SCD- vs vehicle-treated 3xTg mice. The approach, execution and presentation of the research appears well done and, intriguingly, (i) the authors show SCD inhibition reverses AD relevant phenotypes and therefore could be a promising therapeutic target; (ii) the authors suggest new roles for SCD e.g. in regulating microglial activation. To the authors credit, the study is also important as it takes a treatment approach in treating symptomatic animals.

Major points:

1. The data presented in this study is persuasive, specifically regarding reversal of phenotypes in 3x-Tg-AD mice. However, it is not clear whether the phenotype reversal stems from the % reversal of genes altered between wt vs 3x-Tg-AD mice or the "newly changed" genes. This leads to the major concern that this study bases all analyses on a single SCD inhibitor (ab142089). Moreover, the efficacy of this drug for inhibiting SCD is never shown, e.g., by measuring a desaturation index on the S- vs. D-treated mouse brains. Therefore, one cannot be sure that the repeated and surprising (by the authors' own statements) benefits of the SCDi on virtually every phenotype examined in the 3xTG mice can be directly attributed to the MUFA-lowering/SFA-elevating effect of inhibiting SCD in the brain. In short, no in vivo or in vitro pharmacological data are provided for the SCDi. The authors should perform their primary TP analysis of the four genotypes of mice (a la Fig. 1) after one-month infusion of a second SCDi that they believe has closely similar actions to the Abcam compound they used here. The authors should also comment on whether the newly changed genes could be off-target effects.
2. The methods section includes "Experiments were performed using either male C57BL6 mice (Charles River) or female 3xTg-AD mice and their strain controls (below)". Are the mice sex matched throughout the experiments? If so, it would be important to confirm throughout the paper that control and Tg-AD mice were not only age matched (as already stated by the authors) but also sex matched for all experiments given that it is known that fatty acid and lipid metabolism is significantly different between males and females.
3. There appears to be significant variability between animals in the heatmaps presented e.g. in Fig 1. While variability is always expected in vivo, perhaps the authors could comment on this specifically.
4. In Figure 2, were the SCDi treated 3x-Tg-AD mice now not statistically indistinguishable from WT mice?
5. Figure 2 (and others): perhaps the authors could comment on how they view restoration of gene expression versus newly changed genes upon SCDi and attributing rescue to one or both.
6. Attributing/connecting SCD function to MHC-I is an interesting connection, particularly given the synaptic and memory findings. It would be important to follow up on this either with a genetic knockdown or additional SCD inhibitor approach.
7. It would be better if a second APP tg mouse line had also been analyzed to see if the DEGs identified relate to AD mouse models in general or specifically to the 3xTg line.
8. In Fig 1 legend, it is stated "d-e). The most significant biological processes were related to synapses and nervous system development for down-regulated DEGs (d) and immune response and protein metabolism for upregulated DEGs (e). (f) Network plot showing clustering and associations of down-regulated molecular functions: lipid metabolism, membrane channel activity and ion binding." If lipid metabolism was not among the "most significant biological processes", on what basis was the lipid network in (f) pulled out?
9. line 127: Explain better the rationale for jumping on SCD as a key candidate gen worthy of pharmacological inhibition.

10. Clarify interpretation of Fig. 2b: how was this analyzed and what is it conveying?
11. Line 512: What does "ipsilateral hippocampus" mean when the treatment was infused into the lateral ventricles (which freely connect)?
12. In Fig. 2f (legend), was a correction for multiplicity performed in saying that "cutoff for DEG significance was $p < 0.01$ "?
13. Why did $N = 1114$ DEGs in Fig. 1 (8 mos old mice) go down to $N = 777$ DEGs in 9 mos old mice in Fig. 2?
14. Line 161: state the age of the symptomatic mice and describe the time course of their development of symptoms and what the symptoms were.
15. Supp. Fig. 2d-f: on what basis did the authors conclude that a 15 kDa band on a Western blot is "oligomeric amyloid beta"? No blot is shown, and there is no universal definition in the field of a specific 15 kDa A β oligomer seen by WB on a denaturing SDS-PAH+GE gel.
16. Line 273: "Only 5 genes were significantly changed by SCDi in the 3xTG mice" How does this jibe with the earlier statement that SCDi restored 24% of microglial DEGs and newly changed an additional 34 genes?
17. Could the authors clarify the length of time between device implantation and treatment of the mice with SCDi?

Minor points:

- Line 170, 173: these figures should be listed as Fig 3 not Fig 5.
- Labeling of graph bars for Fig 5c (as per Fig 5k)
- Was statistical analysis performed for Fig 5j and 5k?
- Line 385: change "line" to in line with our findings..."
- Fig. 2 legend should also state that (c) is DMSO treatment while (d) is SCDi treatment.
- Line 128: state the identity of the SCD inhibitor used here (with ref).

Reviewer #2:

Remarks to the Author:

Hamilton and colleagues describe a compelling beneficial outcome for SCD inhibitor treatment in 3xTg model of AD, showing improvement in learning and memory as well as rescue of dendritic spine number and complexity. These changes are associated with bulk RNAseq changes in inflammation and synapse development in hippocampus, as well as population changes suggesting reduced activation in purified microglia single cell RNAseq.

The authors spend a lot of time on the various RNAseq results, emphasizing the involvement of lipid metabolism as an important driver of the 3xTg AD model. The RNAseq results do not intuitively lead to this conclusion, but the authors do highlight important lipid gene changes in molecular function GO characterization and also using GSEA. Knowing the action of the SCD inhibitor, and seeing its beneficial effect, it is natural to retrospectively examine what initial lipid disruptions may be modified by such a treatment. However, the way the bulk RNAseq data is presented initially does not make the SCD inhibitor experiment appear the natural next step. The authors previous work (Hamilton et al, Cell Stem Cell, 2015) found a very convincing link between TAG metabolism, lipid saturation, and lipid accumulation in the neurogenic niche of 3xTg. Microarray data implicated SCD-1 and other lipid metabolism genes, and the NSC impairment was ameliorated by SCD inhibition in young mice. These previous experiments were well justified and contextualized, but this same robust reasoning is not as clear in the present manuscript.

Overall, the rescue of cognitive impairment, associated with synaptic density changes, is quite compelling, and it is reasonable to look at the underlying lipid pathways that contribute. The potential for SCDi and other similar targets as novel AD therapeutics is very important. However, the authors need to make significant efforts to both reorganize text and perform additional experiments to clarify the current work.

MAJOR POINTS

- There are no experiments validating the RNAseq results, either bulk or single cell. The authors use GSEA and GO analyses to drive the lipid metabolism link, but do not otherwise confirm that these are in fact modified in their model. The previous work from these authors (Hamilton et al,

Cell Stem Cell 2015) performed imaging mass spectrometry to collect spatial lipid data and showed accumulation of TAG and oleic acid surrounding the lateral ventricles. The authors should show hippocampal lipidomic changes that are relevant to SCD / MUFA metabolism, by either spatial IMS or traditional lipidomics from similar dissected tissue as used for bulk RNAseq. Alternatively, the authors can show significantly changed levels of the genes of interest that contribute to the SCD/MUFA nodes identified with GSEA analyses using IHC, or if that is not possible, RNAscope.

- The authors show apparent strong rescue/restoration of DEGs by SCDi treatment (777 WT-D vs 3xTg-D DEGs, only 62 following SCDi). Likewise, SCDi only induced 9 DEGs in WT mice, but 434 in 3xTg mice. While these results are compelling, I have a difficult time reconciling them with the PCA plot show in Fig. 2b which i) shows a clear shift in WT following SCDi treatment, and ii) does not appear to show 3xTg mice moving closer to WT following SCDi treatment. Can the authors comment? It would also be helpful for the authors to show in Fig. 2c the 777 DEGs across all 4 treatment groups.

- The authors report reduced spine density and dendrite complexity in 3xTg mice that is rescued by SCDi treatment. Are the authors able to detect neuronal loss in these mice? Can it be rescued by SCDi treatment?

- It is not clear why the authors choose to perform single cell RNAseq on microglia, based on their previous results (ie synaptic and lipid related changes). The authors should work on this transition in the text, and include more justification for examining single cell microglia based on their findings to this point (e.g. microglia are known to modify synapses and are also important immune actors in the brain, so we examined whether SCDi treatment was modifying microglia, and if so that this may be leading to the beneficial outcome we observed on dendritic spines....etc).

- The authors state that only 5 genes were significantly changed by SCDi, all MHC class I. They go on to subcluster the microglia, but it is not clear whether these are statistically significantly different, and by what measure

- The single cell analysis again relies on the importance of pathway analysis to draw conclusions, as well as population dynamics between "resting" and "activated" and again the authors claim that SCDi application "restores dendritic spines" but do not provide any staining or other proof that the SCDi microglia are indeed interacting with (and engulfing?) dendritic material. The authors should show staining/localization of microglia to dendritic processes.

- The authors should again show some validation for this single cell experiment.

MINOR POINTS

- When the authors state that a gene is "restored" (e.g. "SCDi restored 322/777 (41%) of DEGs found in 3xTg-D mice..." have these genes been significantly changed by SCDi treatment (ie 3xTg-D vs 3xTg-S) or are just no longer themselves significantly different from WT-D? The description of how the comparisons are done (and which groups are compared) in methods is a bit unclear.

- The authors should include p values for all GO search outcomes, including molecular function (Fig 1f)

- Can the authors explain the rationale for including the ASR microglia subcluster among 'activated' microglia?

- How old are the mice in Fig 3 (9 months?)

- Fig 3a and 3b panels are mislabelled as 5a and 5b

- P values for Fig 4i (immediate early genes increased by SCDi)?

Reviewer #3:

Remarks to the Author:

This article by Hamilton and colleagues investigates fatty acid metabolism in a mouse model of AD. This is a potentially fascinating topic and off the beaten path. A greater mechanistic understanding of such a process could have therapeutic implications down the line. The bulk of the paper relies on several interesting comparisons of symptomatic 3xTg AD model mice with or without an

inhibitor of a fatty acid desaturase which converts saturated FAs into delta9 monounsaturated FAs. The authors show a rescue in Morris water maze, dendritic spine number, and in gene expression changes by bulk, non-cell type specific RNAseq. Assuming that this pharmacology is rigorous and specific, these findings are very important. However, there are many areas of the paper that lack clarity and detail needed to interpret the rigor of the findings. In addition, a mechanistic understanding is largely lacking. A single cell RNAseq dataset focused on microglia and introduced at the end, is not of the highest quality and not well integrated conceptually in the remainder of the paper.

Major comments:

1. The authors never introduce information about the drug, its mechanism of action, or any information about its specificity. A clearer understanding of this is essential to interpreting the paper. Are there potential off target effects? What controls were done to account for this? The drug is introduced in passing in l.127 (and on a side note, and I would encourage writing out of the name upon first use, even if abbreviations are defined in a legend.)

2. Bulk RNA sequencing data is interesting, but some clarification is lacking. It cannot distinguish between differences in cell type abundance vs cell function (indeed, findings of 'increased immune genes' are often tied to microgliosis in different pathologies) – this should be noted. I am somewhat confused by the following statement in the methods: Gene set analysis was accomplished on DEGs using GSEA 4.0.3 [65]. Enriched gene sets were then curated from GSEA using the following keywords; Lipid (lipid, fat, sterol), Inflammation (inflam, cytokine, immun, neutro, antigen, leukocyte, chemotaxis, lymphocyte, interleukin, myeloid, interferon, chemokine), and Synapse (synap, dendri, neurotrans, spine)." The text and main figures seems to suggest that the analysis is 'unbiased'. Could the authors clarify?

3. Figure3: I do not see any quantifications of what I would consider the most relevant metric - surviving newborn neurons (eg. BrdU label retaining 3 weeks after administration, and neuN+) . With data only of proliferation and neuroblast number, I find the analysis rather inconclusive, as new neurons survival and incorporation into circuits is what matters for function. This is a relatively minor point compared to the others.

4. I am not convinced of the quality of the scseq data, and furthermore, it is not clearly related to the remainder of the paper and does not enhance a mechanistic understanding of how FA metabolism impacts AD pathology. Warm dissociation in papain, with the additional time added for multiple centrifugation steps and cd45 labeling would be expected to activate the cells substantially and is not a gold standard approach in the field. The additional labeling step should allow biological replicates to be distinguished bioinformatically to verify reproducibility- I do not see evidence of this in the supplemental data- how many biological replicates does each group represent? Several findings raise concerns: Subcluster 7 DEGS are all expressed in astrocytes including scl1a2, also known as GLT1, a canonical astrocyte gene. If the authors are claiming this is a novel microglial subset, there should be some explanation of why they are expressing astrocyte genes. Cluster 4 is expressing high levels of mag and mog (oligo genes). Mitochondrial DNA threshold is high. Feature plots of counts and mito DNA are not shown to verify that clustering is not driven by differences in quality. Finally, the functional annotations that the authors introduce are extremely confusing and overinterpret the data (see minor point below).

Minor:

The 3xTG AD mouse is never introduced. The three mutations and general course of disease should be described in introduction or results.

Fig 2b- the raw annotations and labels are extremely difficult to read. Please simplify for the reader.

Fig3a-b- please define your abbreviations in the legend.

Fig 3- I interpreted that all tests underwent two way anova with dunnets? Was 3a a repeated

measures anova?

Figure 4- please clarify hippocampal region where Sholl analysis is performed on the the y-axis of plots, or make the legends less confusing.

Figure 4- how many animals per group are represented in the spine quantifications? Please show means per animal next to the total data distribution, and clarify N's in the figure legend.

4i: bulk transcriptomic data on IEGs is not sufficient to make claims about neuronal activity, this should be done in situ with novel exposure followed by c-fos staining, for example.

In addition what do the authors make of the finding that BDNF is so robustly induced by the treatment in WT mice- could suggest alternative mechanisms of spine promotion not related to lipid metabolism per se?

Figure 5- please provide a legend to clarify the treatment groups (c, j, k), please make the figure legend understandable without text. For example, 5d represents which DEGs? Of the microglial subcluster? Or all cells? For 5 g,h,i, cluster numbers should be consistent throughout, including in the violin plots. The multi letter abbreviations are not helpful, as the reader does not know what they mean, and the interpretations may not be correct. For 5b- it might be more fruitful to focus on subcluster analysis of cd45+ only, since that was the goal of the isolation strategy, and as currently presented, the main take home point of the panel is that the selection works.

I.284- what is the rationale for using tSNE instead of umap visualization for the subcluster analysis?

I am somewhat surprised that there are fewer macrophages in the AD model brains – can the authors refer to other literature on the topic that might support this finding?

L173-4- please show the data on swim speed

RESPONSES TO REVIEWER #1

Reviewer #1 (Remarks to the Author):

This work by Hamilton et al., is a compelling, multifaceted study on an important topic, connecting fatty acid metabolism and neurodegeneration in Alzheimers disease. The research presented in this manuscript builds significantly on the groups 2015 work (PMID: 26321199) analyzing transcriptomic differences between WT and symptomatic 3x-Tg-AD mice. It connects the role of SCD and its inhibition in the downstream cellular responses of microglia and neurons to amyloid+tau deposition in a credible model of AD. There are numerous strengths, including the detailed TP and phenotyping, the behavioral readouts, the dendritic quantification and especially the detailed and sometimes novel methods for classifying subpopulations of microglia and their differential TP changes in SCD- vs vehicle-treated 3xTg mice. The approach, execution and presentation of the research appears well done and, intriguingly, (i) the authors show SCD inhibition reverses AD relevant phenotypes and therefore could be a promising therapeutic target; (ii) the authors suggest new roles for SCD e.g. in regulating microglial activation. To the authors credit, the study is also important as it takes a treatment approach in treating symptomatic animals.

Major points:

1. The data presented in this study is persuasive, specifically regarding reversal of phenotypes in 3x-Tg-AD mice. However, it is not clear whether the phenotype reversal stems from the % reversal of genes altered between wt vs 3x-Tg-AD mice or the “newly changed” genes. This leads to the major concern that this study bases all analyses on a single SCD inhibitor (ab142089). Moreover, the efficacy of this drug for inhibiting SCD is never shown, e.g., by measuring a desaturation index on the S- vs. D-treated mouse brains. Therefore, one cannot be sure that the repeated and surprising (by the authors’ own statements) benefits of the SCDi on virtually every phenotype examined in the 3xTG mice can be directly attributed to the MUFA-lowering/SFA-elevating effect of inhibiting SCD in the brain. In short, no in vivo or in vitro pharmacological data are provided for the SCDi. The authors should perform their primary TP analysis of the four genotypes of mice (a la Fig. 1) after one-month infusion of a second SCDi that they believe has closely similar actions to the Abcam compound they used here. The authors should also comment on whether the newly changed genes could be off-target effects.

>>> Response: We have performed **two major series of experiments** to address the concerns about the specificity and mechanism of action of the SCD inhibitor (ab142089).

FIRST: We tested whether a second, commercially available SCD inhibitor that has distinct structural characteristics is able to reproduce key biological and genetic effects that we had observed. We selected the Cayman SCD inhibitor, CAY10566, which has a distinct chemical composition (Ab142089: C₂₀H₂₂ClN₃O₃ vs. CAY10566: C₁₈H₁₇ClFN₅O₂) and physical structure (below), with equal IC₅₀ values of 6.8 nM to facilitate treatment comparison:

ab142089

CAY10566

The CAY10566 (or its vehicle) was loaded into 28-day osmotic pumps and infused into 9-month-old WT and 3xTg mice as originally done for ab142089. At sacrifice, half the brains were used for Golgi staining and dendritic spine quantifications in the DG and CA1 subfields, and half the brains were used for RNA isolation and whole hippocampus RNAseq. Quantification of dendritic spines on Golgi-stained sections revealed virtually identical findings as we had seen with ab142089: specifically, a significant reduction in

spine density in 3xTg mice that was rescued upon inhibitor infusion (see comparison below). **These new spine density data using the CAY10566 inhibitor have been added to the Results (lines 260-264) and new Supplemental Figure 4.**

We then used the second half of these brains for whole hippocampus RNAseq analysis, and found that CAY10566 also has similar genetic effects as the ab142089 inhibitor. When we examined the “New” and “Restored” genes using the ab142089 inhibitor and assessed expression of these same genes in the CAY10566 mice, we observed substantial overlap of changes, especially of the “Restored” genes (below):

Bio-informatic analyses using identical cut-offs showed that the number of “Restored”, “Maintained” and “New” DEGs was virtually identical between our original experiment with ab142089 and new experiment with CAY10566 (comparison below); and notably, the GO Biological Processes for both inhibitors were enriched for synapse- and nervous system growth/development-related gene sets (comparison below). **These new whole hippocampus RNAseq data and bio-informatics analyses with the CAY10566 inhibitor have been added to the Results (lines 176-187) and new Supplemental Figure 2.**

CAY10566

ab142089

SECOND: Regarding the ability of SCDi to inhibit fatty acid desaturation, we have previously shown that this same 28-day SCDi infusion paradigm using ab142089 depletes MUFA-rich triglycerides in the brains of young adult 3xTg mice (Hamilton et al., Cell Stem Cell, 2015, panel reproduced below):

To more specifically assess changes in individual fatty acids species in SCDi-infused mice, we established a collaboration with the laboratory of Dr. Mélanie Plourde (U.Sherbrooke, Canada) to use a sensitive gas chromatography flame-ion detection (GC-FID) method to quantify absolute levels of a panel of 22 medium- and long-chain fatty acids (14-22 carbons in length). Because hippocampal tissues of SCDi-infused mice in the present study had already been processed (for RNAseq, scRNAseq, Golgi staining and tissue sectioning experiments), we analysed microdissections of the nearby subventricular zone that we had conserved from ab142089-infused mice. Lipidomics analysis of the subventricular zone is relevant given that i) SCDi was infused into the CSF (not directly into the hippocampus), and ii) unlike the hippocampus, the SVZ contains high levels of lipid droplets (Hamilton et al., 2015), which provide a readout for the dynamically modulated pool of fatty acids. GC-FID revealed that SCDi infusion in 3xTg mice increased levels of the major SCD substrates (the saturated fatty acids C16:0 and C18:0), as would be predicted, with levels of total saturated fatty acids likewise showing a tendency to be increased (p=0.087). The increased

levels of these SCD substrates, C16:0 and C18:0, was not accompanied by a decrease in the SCD products, C16:1 and C18:1; this is not surprising given that these experiments are chronic rather than acute (i.e., 1 month infusions) and have complex biological consequences (i.e., changes in spine number, dendritic length and microglial activity). GC-FID also identified one longer chain fatty acid, C22:6 (DHA), that was significantly increased by SCDi in 3xTg mice. Interestingly, the concentrations of both total fatty acids and total saturated fatty acids were significantly reduced in 3xTg-D mice (versus their WT-D counterparts) and were restored in 3xTg-S mice. Overall, these lipidomic analyses support that SCDi is modulating its molecular target and modifying the overall brain fatty acid composition. **These new data have been added to the Results (lines 147-150) and to new Supplemental Figure 1a.**

Fatty acid (mg/g)	Strain and treatment				P value
	WT-DMSO	WT-SCDi	3xTg-DMSO	3xTg-SCDi	
C14:0	0.045 ± 0.025	0.072 ± 0.033	0.051 ± 0.030	0.071 ± 0.016	ns
C15:0	0.012 ± 0.012	0.053 ± 0.009	0.035 ± 0.004	0.052 ± 0.006	ns
C16:0	7.179 ± 0.616	8.448 ± 0.957	6.494 ± 0.251	8.160 ± 1.043	WT _D vs WT _S : 0.007 T _{GD} vs T _{GS} : 0.001
C16:1 t	0.040 ± 0.021	0.055 ± 0.007	0.044 ± 0.004	0.069 ± 0.001	ns
C16:1	0.131 ± 0.022	0.140 ± 0.015	0.118 ± 0.018	0.147 ± 0.018	ns
C18:0	9.226 ± 0.743	9.955 ± 0.717	8.456 ± 0.291	9.469 ± 1.214	WT _D vs WT _S : 0.124 WT _D vs T _{GD} : 0.129 T _{GD} vs T _{GS} : 0.046
C18:1 n-9	6.026 ± 0.912	5.888 ± 0.387	4.855 ± 0.770	5.683 ± 0.696	WT _D vs T _{GD} : 0.021 T _{GD} vs T _{GS} : 0.103
C18:1 n-7	1.400 ± 0.177	1.500 ± 0.114	1.247 ± 0.117	1.444 ± 0.234	ns
C18:2 t	0.573 ± 0.152	0.493 ± 0.061	0.416 ± 0.078	0.466 ± 0.061	ns
C18:2 n-6	0.290 ± 0.033	0.328 ± 0.027	0.303 ± 0.018	0.343 ± 0.053	ns
C20:0	0.104 ± 0.013	0.110 ± 0.007	0.122 ± 0.008	0.112 ± 0.006	ns
C20:1	0.559 ± 0.113	0.473 ± 0.056	0.511 ± 0.062	0.479 ± 0.024	ns
C20:3 n-6 DGLA	0.143 ± 0.023	0.147 ± 0.012	0.128 ± 0.013	0.155 ± 0.027	ns
C20:4 n-6	4.642 ± 0.498	4.976 ± 0.368	4.086 ± 0.476	4.958 ± 0.869	T _{GD} vs T _{GS} : 0.086
C22:3	1.445 ± 0.235	1.479 ± 0.125	1.187 ± 0.161	1.393 ± 0.207	ns
C24:1	0.162 ± 0.085	0.076 ± 0.044	0.053 ± 0.053	0.000 ± 0.000	ns
C22:5 n-3 DPA	0.021 ± 0.021	0.043 ± 0.015	0.039 ± 0.020	0.048 ± 0.024	ns
C22:6 n-3 DHA	4.883 ± 0.550	5.209 ± 0.391	4.207 ± 0.544	5.489 ± 1.076	T _{GD} vs T _{GS} : 0.012
ΣSFA	20.31 ± 2.763	20.81 ± 1.490	17.19 ± 2.096	20.67 ± 3.283	WT _D vs T _{GD} : 0.124 T _{GD} vs T _{GS} : 0.087
ΣUFA	16.57 ± 1.332	18.65 ± 1.659	15.16 ± 0.371	17.88 ± 2.259	ns
ΣFA	36.88 ± 4.081	39.46 ± 2.828	32.35 ± 2.460	38.56 ± 5.527	WT _D vs T _{GD} : 0.027 T _{GD} vs T _{GS} : 0.003

2. The methods section includes “Experiments were performed using either male C57BL6 mice (Charles River) or female 3xTg-AD mice and their strain controls (below)”. Are the mice sex matched throughout the experiments? If so, it would be important to confirm throughout the paper that control and Tg-AD mice were not only age matched (as already stated by the authors) but also sex matched for all experiments given that it is known that fatty acid and lipid metabolism is significantly different between males and females.

>>> **Response:** This was a text error: *male C57BL6 mice (Charles River)* were not used in the experiments included in this study. Female 3xTg-AD mice and their strain controls (Jax mice) were used throughout the study and were indeed precisely age-matched. The Methods have been modified accordingly (lines 759-760).

3. There appears to be significant variability between animals in the heatmaps presented e.g. in Fig 1. While variability is always expected *in vivo*, perhaps the authors could comment on this specifically.

>>> **Response:** In addition to the *in vivo* context and usual technical factors (precision of brain microdissections, RNA extractions, etc), the inter-animal variability seen in Figure 1 is likely attributable to the fact that 8-month-old mice carrying disease-causing AD mutations do not develop their disease pathologies with completely identical time courses. Given these factors, we are pleased overall with the reproducibility we obtained in these experiments.

4. In Figure 2, were the SCDi treated 3x-Tg-AD mice now not statistically indistinguishable from WT mice?

>>> **Response:** The answer to this question is No, for two reasons. First, although SCDi treatment in 3xTg mice eliminated 322/777 DEGs that were present between WT-D and 3xTg-D, it also stimulated 567 new changes in gene expression versus WT-D. Second, we have conducted additional bio-informatics to further analyze the 322/777 WT-D/3xTg-D DEGs that were eliminated in 3xTg-S mice. Importantly, this confirms that the WT-D versus 3xTg-D difference in expression of the “Restored” genes was substantially reduced (by an average of 42%), but also highlights that these genes are not completely normalized to WT-D levels (figure below).

The text of the Results has been revised accordingly (lines 161-169) and the above panel has been added to new Supplementary Figure 1b.

5. Figure 2 (and others): perhaps the authors could comment on how they view restoration of gene expression versus newly changed genes upon SCDi and attributing rescue to one or both.

>>> **Response:** It is not possible to conclude *definitively* on the functional contributions of the “Restored” versus “New” DEGs. Indeed, GO analysis showed that both “Restored” and “New” DEGs are enriched in gene sets relating to synapses and neuronal growth/development, suggesting both are contributing to similar processes. However, as described in point 1 above, our new data shows that two different SCDi (Ab142089 and CAY10566), which both restored 3xTg dendritic spines, are particularly similar with regards to their effects on “Restored” genes. This suggests that Restored genes may be more centrally involved in mediating SCDi effects. **The text of the Results has now been modified to include this observation (lines 176-187).**

6. Attributing/connecting SCD function to MHC-I is an interesting connection, particularly given the synaptic and memory findings. It would be important to follow up on this either with a genetic knockdown or additional SCD inhibitor approach.

AND

7. It would be better if a second APP tg mouse line had also been analyzed to see if the DEGs identified relate to AD mouse models in general or specifically to the 3xTg line.

>>> **Response:** To address these two related points, we have conducted new SCD inhibition experiments and performed bio-informatic analyses of previously published datasets from other Alzheimer mouse models.

FIRST: We have used an additional SCD inhibition approach to further strengthen the link between SCD and MHC-1 expression in microglia. To do so, we established a collaboration with the laboratory of Dr. Catherine Mounier (UQAM, Canada) to study microglia from *Scd1* KO mice. Neonatal microglial cultures were established for 11-13 days from WT and *Scd1*-KO (-/-) littermates, then treated for 6hrs with either lipopolysaccharide (LPS) or Vehicle and analyzed by quantitative PCR (qPCR). As expected, qPCR analysis confirmed that LPS significantly increased mRNA expression of TNF α and IL1 β (two main markers of microglia activation). Importantly, this upregulation of TNF α and IL1 β was attenuated in *Scd1*-KO microglial cultures and in WT cultures treated with SCDi, demonstrating a critical role of *Scd* during inflammatory activation of microglia. Furthermore, expression of H2-D1 and H2-K1 was likewise increased following LPS exposure and this was blunted in both *Scd1*-KO microglia and in WT microglia treated with SCDi, demonstrating that inflammation-induced upregulation of MHC-I genes is largely mediated by microglial *Scd* expression. Since H2-D1 and H2-K1 were the main transcriptomic targets of SCDi in 3xTg microglia (Fig. 5i), these data are in line with the idea that *Scd* plays an essential role in microglia activation/response and that this may be mediated via MHC-I gene expression, specifically H2-D1 and H2-K1. **This new experiment has been added to the Results (lines 333-345) and to Figure 5.**

SECOND: We asked whether the 3xTg-associated DEGs identified in the present study (especially the SCD-regulated MHC-1 molecules, H2-D1 and H2-K1) are relevant to other AD mouse models as well. To do so, we overlapped our WT-D/3xTg-D microglial DEGs (scRNAseq) to the DEG lists from four other published datasets: Sala Frigerio et al (Cell Reports, 2019; scRNAseq of FACS-isolated microglia from App^{NL-G-F} KI mice), Baik et al (Cell Metabolism, 2019; bulk RNAseq of cultured microglia from 5xFAD mice), Sierksma et al (EMBO Mol Med, 2020; scRNAseq of FACS-isolated microglia from APP/PS1 mice), and Xia et al (GEO accession# GSE158153 made public 2021; bulk RNAseq of microglia from APP KI mice carrying 3 FAD mutations). This analysis revealed an overlap of 12 DEGs across all AD models (Abcd2, Bank1, Ccl6, Ctsl, Fam102b, Filip1l, **H2-D1**, **H2-K1**, Nrp1, Pnp, Rplp1, Rps13) (figure below). Since H2-D1 and H2-K1 were among the 12 shared DEGs, this suggests that SCDi regulates a microglial mechanism that is common across multiple AD mouse models. **This new analysis has been added to the Results (lines 323-332) and to Figure 5j.**

8. In Fig 1 legend, it is stated “d-e). The most significant biological processes were related to synapses and nervous system development for down-regulated DEGs (d) and immune response and protein metabolism for upregulated DEGs (e). (f) Network plot showing clustering and associations of down-regulated molecular functions: lipid metabolism, membrane channel activity and ion binding.” If lipid metabolism was not among the “most significant biological processes”, on what basis was the lipid network in (f) pulled out?

>>> **Response:** This was not clearly stated in our original text. The lipid network was pulled out as one of the significantly enriched GO Molecular Functions gene sets. **We have modified the description to improve its clarity (lines 104-115).**

9. line 127: Explain better the rationale for jumping on SCD as a key candidate gen worthy of pharmacological inhibition.

>>> **Response:** We agree that our previous text suggested that our interest in SCD arose mainly from the present GO enrichment analyses. **We have now modified the Results section (lines 140-144)** to clarify that our interest in SCD as a potential therapeutic target originated from our prior study, which showed SCD inhibition restores neural stem cell activity in pre-symptomatic 3xTg mice (Hamilton et al., 2015), and that this was further supported by the transcriptomic alterations in fatty acid metabolism pathways seen in Figure 1 of the present study.

10. Clarify interpretation of Fig. 2b: how was this analyzed and what is it conveying?

>>> **Response:** The PCA plot (Fig. 2b in the revised Fig. 2) provides a multi-dimensional visualization of transcriptomic differences between the 4 experimental groups. In this case, we used all genes to generate the PCA plot. **We have now revised the Results section (lines 151-160) to specify more clearly what it conveys, and how this is consistent with the changes in numbers of DEGs:** “Principle component analysis (PCA) using the entire transcriptome (all genes) showed that the mice clustered according to their experimental groups, that SCDi had a larger impact on overall gene expression in 3xTg than WT mice, and that SCDi-infused WT and 3xTg mice both remained transcriptomically distinct from the DMSO-infused WT mice (Fig. 2b). Consistent with these observations, SCDi treatment resulted in more statistically significant DEGs in 3xTg mice (3xTg-D vs 3xTg-S, 434 DEGs, Supplemental Data File 2) than in WT mice (WT-D vs WT-S, 9 DEGs, Supplemental Data File 2), and the total number of DEGs between WT-D and 3xTg-D groups (777 DEGs) and between WT-D and 3xTg-S groups (1022 DEGs) were similarly high.”

11. Line 512: What does “ipsilateral hippocampus” mean when the treatment was infused into the lateral ventricles (which freely connect)?

>>> **Response:** “Ipsilateral hippocampus” refers to the side on which the osmotic pump was implanted. We systematically used the same side from all animals to minimize inter-animal differences in SCDi diffusion and/or local side-effects of the pump canula implantation. To avoid confusion, we have removed the term ipsilateral and now stated “followed by rapid dissection of the hippocampus from the pump-implanted side” (line 867).

12. In Fig. 2f (legend), was a correction for multiplicity performed in saying that “cutoff for DEG significance was $p < 0.01$ ”?

>>> **Response:** The original figure legend was indeed confusing and has now been modified for clarity. As detailed in the Methods, DEG lists were identified using a cut-off of $p \leq 0.01$ for bulk RNAseq experiments, $p\text{-adj} \leq 0.05$ for single cell RNAseq, and a false discovery rate (FDR) of ≤ 0.05 was applied for GO enrichment analyses. In Fig.2, we show only the 20 most enriched gene sets that were above the FDR cut-off.

13. Why did $N = 1114$ DEGs in Fig. 1 (8 mos old mice) go down to $N = 777$ DEGs in 9 mos old mice in Fig. 2?

>>> **Response:** It is likely that the decreased number of DEGs is not because the mice are one month older, but rather because all mice in Fig. 2 were implanted with an osmotic pump canula into the brain. For example, any inflammatory response due to pump implantation would be predicted to mask WT/3xTg DEGs in immune-related gene expression.

14. Line 161: state the age of the symptomatic mice and describe the time course of their development of symptoms and what the symptoms were.

>>> **Response:** We have now briefly described pathology development in these mice in both the Results (lines 98-100) and Methods (lines 744-760) sections

15. Supp. Fig. 2d-f: on what basis did the authors conclude that a 15 kDa band on a Western blot is “oligomeric amyloid beta”? No blot is shown, and there is no universal definition in the field of a specific 15 kDa A β oligomer seen by WB on a denaturing SDS-PAH+GE gel.

>>> **Response:** This was an error, the band is at 12KDa not 15KDa, this has been rectified in the figure.

16. Line 273: “Only 5 genes were significantly changed by SCDi in the 3xTG mice” How does this jibe with the earlier statement that SCDi restored 24% of microglial DEGs and newly changed an additional 34 genes?

>>> **Response:** These are related but distinct statistical comparisons. “Only 5 genes were significantly changed by SCDi in the 3xTG mice” refers to a direct comparison of microglia from 3xTg-D versus 3xTg-S. “SCDi restored 24% of microglial DEGs and newly changed an additional 34 genes” refers to that when overlapping WT-D/3xTg-D (DEG list #1) with WT-D/3xTg-S (DEG list #2), 24% of the genes from DEG list #1 disappeared from DEG list #2, and 34 new genes appear in DEG list #2 that were not present in DEG list #1. Both of these statistical comparisons are informative but they are statistically distinct because the FDR/p(adj) of individual genes is influenced by the expression levels of all transcripts in each experimental group. Importantly, however, these comparisons are related: specifically, one can see that H2-D1 and H2-K1 (the most highly expressed of the 5 genes that significantly changed between 3xTg-D and 3xTg-S, Fig. 5i) were also DEGs between WT-D/3xTg-D and do indeed appear on the list of genes “Restored” by SCDi (Fig. 5g). Thus, a major genetic effect of SCDi in microglia is reversing the upregulation of these MHC-I genes. **The text of the Results has been modified to clarify this (lines 310-322).**

17. Could the authors clarify the length of time between device implantation and treatment of the mice with SCDi?

>>> **Response:** We have now specified this in the Methods (lines 788-790): “The 28-day Alzet osmotic pumps used in these experiments (0.11 µl/h infusion rate, model 1004; Durect) were primed for 48 hrs and begin pumping immediately when implanted.”

Minor points:

-Line 170, 173: these figures should be listed as Fig 3 not Fig 5.

>>> **Response:** This has been fixed

-Labeling of graph bars for Fig 5c (as per Fig 5k)

>>> **Response:** This has been fixed

-Was statistical analysis performed for Fig 5j and 5k?

>>> **Response:** Statistical analysis cannot be performed for these panels as each bar represents a single pooled calculation, i.e., for each of the experimental groups (WT-D, WT-S, 3xTg-D, 3xTg-S), 10 000 cells were pooled together from N=4 animals. Due to limitations in the number of multiplexing labels, the multiplexing was performed for the pooled sample from each experimental group (not for each animal).

-Line 385: change “line” to in line with our findings...”

>>> **Response:** This has been fixed

-Fig. 2 legend should also state that (c) is DMSO treatment while (d) is SCDi treatment.

>>> **Response:** Figure has been modified.

-Line 128: state the identity of the SCD inhibitor used here (with ref).

>>> **Response:** Inhibitor information has been added to the Methods (lines 776-780).

RESPONSES TO REVIEWER #2

Reviewer #2 (Remarks to the Author):

Hamilton and colleagues describe a compelling beneficial outcome for SCD inhibitor treatment in 3xTg model of AD, showing improvement in learning and memory as well as rescue of dendritic spine number and complexity. These changes are associated with bulk RNAseq changes in inflammation and synapse development in hippocampus, as well as population changes suggesting reduced activation in purified microglia single cell RNAseq.

The authors spend a lot of time on the various RNAseq results, emphasizing the involvement of lipid metabolism as an important driver of the 3xTg AD model. The RNAseq results do not intuitively lead to this conclusion, but the authors do highlight important lipid gene changes in molecular function GO characterization and also using GSEA. Knowing the action of the SCD inhibitor, and seeing its beneficial effect, it is natural to retrospectively examine what initial lipid disruptions may be modified by such a treatment. However, the way the bulk RNAseq data is presented initially does not make the SCD inhibitor experiment appear the natural next step. The authors previous work (Hamilton et al, Cell Stem Cell, 2015) found a very convincing link between TAG metabolism, lipid saturation, and lipid accumulation in the neurogenic niche of 3xTg. Microarray data implicated SCD-1 and other lipid metabolism genes, and the NSC impairment was ameliorated by SCD inhibition in young mice. These previous experiments were well justified and contextualized, but this same robust reasoning is not as clear in the present manuscript.

Overall, the rescue of cognitive impairment, associated with synaptic density changes, is quite compelling, and it is reasonable to look at the underlying lipid pathways that contribute. The potential for SCDi and other similar targets as novel AD therapeutics is very important. However, the authors need to make significant efforts to both reorganize text and perform additional experiments to clarify the current work.

MAJOR POINTS

- There are no experiments validating the RNAseq results, either bulk or single cell. The authors use GSEA and GO analyses to drive the lipid metabolism link, but do not otherwise confirm that these are in fact modified in their model. The previous work from these authors (Hamilton et al, Cell Stem Cell 2015) performed imaging mass spectrometry to collect spatial lipid data and showed accumulation of TAG and oleic acid surrounding the lateral ventricles. The authors should show hippocampal lipidomic changes that are relevant to SCD / MUFA metabolism, by either spatial IMS or traditional lipidomics from similar dissected tissue as used for bulk RNAseq. Alternatively, the authors can show significantly changed levels of the genes of interest that contribute to the SCD/MUFA nodes identified with GSEA analyses using IHC, or if that is not possible, RNAscope.

>>> Response: See our detailed response to Reviewer #1 (point 1). To summarize, **1.** we have now replicated key gene expression and biological findings using a second, distinct SCD inhibitor, confirming that it has overlapping genetic effects (especially on the “Restored” genes) and that it similarly rescues the loss of dendritic spines; **2.** We have used GC-FID to document the effects of SCDi on the overall fatty acid composition of the nearby subventricular zone, showing that 3xTg mice have diminished levels of saturated fatty acids that are indeed rescued by SCDi infusion. We had initially attempted to use MALDI spatial IMS on sections of the hippocampus for this lipidomic analysis (as suggested by Reviewer 2), but triglyceride and free fatty acid levels were too low to obtain reliable data. For that reason, we eventually used the nearby subventricular zone, which (unlike the hippocampus) contains large numbers of lipid droplets that represent the dynamically modulated pool of fatty acids, enabling assessment of inhibitor-induced changes in fatty acid composition. In ongoing studies, we are using MALDI spatial IMS to identify the species of complex lipids that are modulated by SCDi in the hippocampus, potentially providing insight into the downstream lipid pathways modulated by SCDi; however, the latter goes beyond the scope of verifying target engagement by the inhibitor and is an entire study unto itself.

- The authors show apparent strong rescue/restoration of DEGs by SCDi treatment (777 WT-D vs 3xTg-D DEGs, only 62 following SCDi). Likewise, SCDi only induced 9 DEGs in WT mice, but 434 in 3xTg mice. While these results are compelling, I have a difficult time reconciling them with the PCA plot shown in Fig. 2b which i) shows a clear shift in WT following SCDi treatment, and ii) does not appear to show 3xTg mice moving closer to WT following SCDi treatment. Can the authors comment? It would also be helpful for the authors to show in Fig. 2c the 777 DEGs across all 4 treatment groups.

>>> **Response:** Please see our detailed response to Reviewer 1 (point 4). Additionally, it should be noted that numbers of DEGs are based on an arbitrarily selected P-value cutoff, but genes that do not pass the selected cut-off do not necessarily have equal expression levels. The PCA plot was performed using all genes in the transcriptome, not simply the DEGs, explaining why there is “a clear shift in WT following SCDi treatment” even though SCDi treatment only resulted in 9 DEGs in WT mice. For the same reason, the PCA plot “does not appear to show 3xTg mice moving closer to WT following SCDi treatment” because it is not based only on the “Restored” genes, but also includes the “New” genes (newly changed versus WT) and the rest of the transcriptome. **We have reworded this section of the Results for clarity (lines 151-160), and as requested, we have also added the full heatmap of the 777 WT-D/3xTg-D DEGs (Figure 2c).**

- The authors report reduced spine density and dendrite complexity in 3xTg mice that is rescued by SCDi treatment. Are the authors able to detect neuronal loss in these mice? Can it be rescued by SCDi treatment?

>>> **Response:** Neuronal loss is not reported to be a significant feature in 3xTg mice. Indeed, 3xTg mice are a relatively mild amyloidogenic Alzheimer model, in that they develop amyloid plaques later than many other AD mouse models and neurofibrillary tangles only begin appearing at timepoints later than those examined in the present study (see <https://www.alzforum.org/research-models/3xtg> for summary of pathology development). To examine this in our own hands, we performed Nissl staining and quantified neuronal density and pyramidal and molecular layer thickness measurements across genotypes and treatments. This revealed no evidence for an effect of SCDi on these parameters. **These new analyses have been added to the Results (lines 234-235) and Figure 3 (k-n).**

- It is not clear why the authors choose to perform single cell RNAseq on microglia, based on their previous results (ie synaptic and lipid related changes). The authors should work on this transition in the text, and include more justification for examining single cell microglia based on their findings to this point (e.g. microglia are known to modify synapses and are also important immune actors in the brain, so we examined whether SCDi treatment was modifying microglia, and if so that this may be leading to the beneficial outcome we observed on dendritic spines....etc).

>>> **Response:** We have followed the Reviewer’s suggestion for clarifying the logic of studying microglia, adding a paragraph (lines 280-290) to clarify the rationale immediately before the scRNAseq, as follows: “Microglia, the resident immune cells of the central nervous system, are crucial for normal synapse development, support and elimination [28, 29]. They also disproportionately express AD risk genes [30] and are implicated in AD-associated synapse loss [29, 31], making them central players during AD pathogenesis [32, 33]. Given that SCDi “restored” 39% of immune DEGs (94/238) and stimulated 159 “new” immune-related DEGs in the 3xTg hippocampus (Supplemental Data File 2), we asked whether SCDi-mediated changes in microglial activity contributes to its beneficial effects on dendritic spines.”

- The authors state that only 5 genes were significantly changed by SCDi, all MHC class I. They go on to subcluster the microglia, but it is not clear whether these are statistically significantly different, and by what measure

>>> **Response:** We confirm that, yes, these 5 MHC-I genes are the only DEGs significantly changed by SCDi in 3xTg microglia (p-values have now been added to the figure legend). We have also added additional panels that show the relative expression of these genes across experimental groups (**Fig. 5i**) and across microglial subclusters (**Fig. 6g**), as well as the effects of genotype and treatment on expression of the H2-D1 and H2-K1 in the AR/AIR lineage and in the ASR lineage (**Fig. 6h,i** and **Supplemental Fig. 7a,b**).

- *The single cell analysis again relies on the importance of pathway analysis to draw conclusions, as well as population dynamics between “resting” and “activated” and again the authors claim that SCDi application “restores dendritic spines” but do not provide any staining or other proof that the SCDi microglia are indeed interacting with (and engulfing?) dendritic material. The authors should show staining/localization of microglia to dendritic processes.*

>>> **Response:** As requested, we have performed a new, immunostaining analysis of Iba1+ microglia and PSD95+ postsynaptic densities (new Figure 5 a,b) in the CA1 zone, where we had detected an SCDi-mediated rescue of dendritic spines. Blinded confocal analyses revealed that Iba1-PSD95 contact points were significantly reduced in the 3xTg-D mice but not in the 3xTg-S mice. Thus, at this 10-month-old timepoint, SCDi treatment increases microglia-synapse interactions to wildtype levels. We do not speculate on what an SCDi-mediated increase in microglia-synapse interactions might mean (slowed microglial engulfment of spines? Enhanced microglial synaptic support (Parkhurst et al., Cell, 2013)?), but these data provide support for the subsequent single cell analysis of microglia. **These new data have been added to Fig. 5 a,b, and further strengthen the rationale for the subsequent single cell analysis of SCDi effects on microglia.**

- *The authors should again show some validation for this single cell experiment.*

>>> **Response:** Please see our detailed responses to Reviewer 1 (points 6 and 7). Briefly, our single cell experiment identified the MHC-I genes, H2-D1 and H2-K1, as the two genes most prominently regulated by SCDi in 3xTg microglia. We have now **1.** performed in vitro experiments using microglial cultures treated with SCDi or derived from Scd1-KO mice, both showing that Scd is critical for inflammatory activation of microglia and does indeed regulate MHC-I genes in microglia; and **2.** bio-informatically compared our single cell RNAseq microglial DEGs with publicly available datasets from 4 other AD mouse models, showing that H2-D1 and H2-K1 are among the core set of 12 genes that overlapped across all 5 AD models.

MINOR POINTS

- *When the authors state that a gene is “restored” (e.g. “SCDi restored 322/777 (41%) of DEGs found in 3xTg-D mice...” have these genes been significantly changed by SCDi treatment (ie 3xTg-D vs 3xTg-S) or are just no longer themselves significantly different from WT-D? The description of how the comparisons are done (and which groups are compared) in methods is a bit unclear.*

>>> **Response:** Please see our detailed response to Reviewer 1 (point 4), where we have provided a more detailed analysis of the “Restored” genes. Briefly, we show that the expression level of these genes in 3xTg-S mice does not return to WT levels, but the difference in their expression is reduced by an average of 42% (new Supplementary Figure 1b). As requested, we have also clarified the explanation of how the comparisons are made (lines 161-169).

- *The authors should include p values for all GO search outcomes, including molecular function (Fig 1f)*

>>> **Response:** P-values/FDR have now been added for all GO searches outcomes, in either the figures themselves, the Results, or in Supplemental Data File 1.

- Can the authors explain the rationale for including the ASR microglia subcluster among 'activated' microglia?

>>> **Response:** All three microglial subtypes that were annotated as “activated” (AR, AIR, ASR) exhibit increased pseudotime transition by trajectory inference (below left) and elevated nCount (number of RNAs/cell, below right), both of which are indicative of a more active cellular state. The AR and AIR subtypes were annotated as such based on their gene profiles that aligned with previous articles, as detailed in the Results/Methods, while the ASR appears to represent a novel microglial subcluster. We have clarified this rationale in the Results/Methods. **These data below have been added to a new Fig. 6 (c,d).**

- How old are the mice in Fig 3 (9 months?)

>>> **Response:** Yes, the mice are 9 months at the start of the experiment, this is clarified in the text.

- Fig 3a and 3b panels are mislabelled as 5a and 5b

>>> **Response:** Panels are re-labelled.

- P values for Fig 4i (immediate early genes increased by SCDi)?

>>> **Response:** We have added the p-values to the Results and Legend.

REVIEWER COMMENTS – REVIEWER #3

Reviewer #3 (Remarks to the Author):

This article by Hamilton and colleagues investigates fatty acid metabolism in a mouse model of AD. This is a potentially fascinating topic and off the beaten path. A greater mechanistic understanding of such a process could have therapeutic implications down the line. The bulk of the paper relies on several interesting comparisons of symptomatic 3xTG AD model mice with or without an inhibitor of a fatty acid desaturase which converts saturated FAs into delta9 monounsaturated FAs. The authors show a rescue in Morris water maze, dendritic spine number, and in gene expression changes by bulk, non-cell type specific RNAseq. Assuming that this pharmacology is rigorous and specific, these findings are very important. However, there are many areas of the paper that lack clarity and detail needed to interpret the rigor of the findings. In addition, a mechanistic understanding is largely lacking. A single cell RNAseq dataset focused on microglia and introduced at the end, is not of the highest quality and not well integrated conceptually in the remainder of the paper.

Major comments:

1. The authors never introduce information about the drug, its mechanism of action, or any information about its specificity. A clearer understanding of this is essential to interpreting the paper. Are there potential off target effects? What controls were done to account for this? The drug is introduced in passing in l.127 (and on a side note, and I would encourage writing out of the name upon first use, even if abbreviations are defined in a legend.)

>>> Response: Regarding inhibitor *mechanism of action* and *specificity*, please see our detailed responses to Reviewer #1 (points 1 and points 6/7). Briefly: **1.** We have now replicated key gene expression and biological findings using a second molecularly distinct SCD inhibitor, demonstrating that it has overlapping genetic effects (especially on the bulk RNAseq “Restored” genes) and likewise rescues the loss of dendritic spines (new Supplemental Figures 2 and 4); **2.** Since Scd mediates delta-9 mono-unsaturation of C16:0 and C18:0, we have used GC-FID to assess brain fatty acid composition, showing that 3xTg mice have diminished levels of saturated fatty acids that are indeed rescued by SCDi infusion (new Supplemental Figure 1); **3.** We have added experiments using primary cultures of microglia, showing that inflammatory activation of microglia and their upregulation of MHC1 components is abrogated using multiple SCD inhibition approaches (SCDi or SCD1-KO microglia) (new Figure 5k-n). As requested, we have also provided more detailed information in the Methods (lines 776-680) about the SCD inhibitors that we have used in this study (both the original Ab142089 and the newly added CAY10566).

2. Bulk RNA sequencing data is interesting, but some clarification is lacking. It cannot distinguish between differences in cell type abundance vs cell function (indeed, findings of ‘increased immune genes’ are often tied to microgliosis in different pathologies) – this should be noted. I am somewhat confused by the following statement in the methods: Gene set analysis was accomplished on DEGs using GSEA 4.0.3 [65]. Enriched gene sets were then curated from GSEA using the following keywords; Lipid (lipid, fat, sterol), Inflammation (inflam, cytokine, immun, neutro, antigen, leukocyte, chemotaxis, lymphocyte, interleukin, myeloid, interferon, chemokine), and Synapse (synap, dendri, neurotrans, spine).” The text and main figures seems to suggest that the analysis is ‘unbiased’. Could the authors clarify?

>>> Response: Thank you for this comment, we have added additional technical clarifications as requested, and have specified in the Results that the changes in bulk RNAseq data might be mediated by changes in cell type abundance and/or cell function (lines 131-132). Regarding the GSEA analysis, this was done in order to identify the lipid, immune and synapse genes within our DEG list. To do so, our goal was to use an objective approach (via GSEA) rather than to use a subjective hand-curation of the individual genes. Specifically: **1.** We used the statistically significant DEGs ($p < 0.01$) to perform GSEA, revealing all gene sets that have DEG enrichment; **2.** We then used the indicated keywords to identify the GSEA gene sets were specifically related to Lipid (lipid, fat, sterol), Immune (inflam, cytokine, immun, neutro, antigen, leukocyte, chemotaxis, lymphocyte, interleukin, myeloid, interferon, chemokine), and Synapse (synap,

dendri, neurotrans, spine); 3. The DEGs present within these enriched Lipid, Immune and Synapse gene sets represent the lipid, immune and synapse genes within our DEG list. **We have re-written the description of this step in the Results (lines 120-122) and Methods (lines 856-861) to be more clear.**

3. Figure 3: I do not see any quantifications of what I would consider the most relevant metric - surviving newborn neurons (eg. BrdU label retaining 3 weeks after administration, and neuN+). With data only of proliferation and neuroblast number, I find the analysis rather inconclusive, as new neurons survival and incorporation into circuits is what matters for function. This is a relatively minor point compared to the others.

>>> Response: We agree with the Reviewer that we have not excluded an effect of SCDi on the downstream newborn neurons, and should therefore not use the term “adult neurogenesis”. In reality, our main focus was on the upstream stages of the neurogenic pathway (neural stem/progenitors and neuroblasts) because these were the cell populations that we had previously found to be rescued upon SCDi treatment in young adult 3xTg mice (Hamilton et al., Cell Stem Cell, 2015). We therefore repeated the same quantifications we had used in our prior study (Ki67+ total proliferating cells, Ki67+GFAP+ proliferating NSCs, Ki67+/DCX+ proliferating neuroblasts and DCX+ total neuroblasts). We have now modified the text everywhere to more accurately specify that SCDi did not impact “neural stem/progenitor proliferation and neuroblast numbers”.

4. I am not convinced of the quality of the scseq data, and furthermore, it is not clearly related to the remainder of the paper and does not enhance a mechanistic understanding of how FA metabolism impacts AD pathology. Warm dissociation in papain, with the additional time added for multiple centrifugation steps and cd45 labeling would be expected to activate the cells substantially and is not a gold standard approach in the field. The additional labeling step should allow biological replicates to be distinguished bioinformatically to verify reproducibility- I do not see evidence of this in the supplemental data- how many biological replicates does each group represent? Several findings raise concerns: Subcluster 7 DEGs are all expressed in astrocytes including scl1a2, also known as GLT1, a canonical astrocyte gene. If the authors are claiming this is a novel microglial subset, there should be some explanation of why they are expressing astrocyte genes. Cluster 4 is expressing high levels of mag and mog (oligo genes). Mitochondrial DNA threshold is high. Feature plots of counts and mito DNA are not shown to verify that clustering is not driven by differences in quality. Finally, the functional annotations that the authors introduce are extremely confusing and overinterpret the data (see minor point below).

>>> Response: Regarding the specific concerns:

Subcluster 7: We apologize for the confusion. The supplementary file contains multiple tabs, and it appears that the Reviewer was actually looking at “Cluster” 7 of the total cell population, rather than the microglial “Subclusters”. Therefore, yes, the DEGs in Cluster 7 are indeed canonical astrocyte genes because this cluster was actually annotated as being astrocytes. **We have modified the labelling and specified the cell annotations within the file to make it easier to follow.**

Cluster 4: Here, it appears the reviewer is indeed referring to “Subcluster” 4, which does express certain genes typically associated with oligodendrocytes (as we had indicated in the text). We do not have an explanation for this subcluster at the moment, but microglia expressing oligodendrocyte transcripts has been reported previously in the literature (Solga et al., 2015, Glia) and may have been picked up here due to the sequencing depth we achieved.

Mitochondrial DNA threshold: Our original description was unfortunately misleading. There are wide-ranging differences in mitochondrial content cut-offs used for quality control because there are considerable cell type-specific differences in mitochondrial content (as clearly illustrated in Supplemental Figure 5d, and mentioned in <https://pubmed.ncbi.nlm.nih.gov/30094294/>). We had used an initial cut-off of 27% for all cells in our scRNAseq in order to avoid loss of valid data points for other cell types, but in fact

the mitochondrial content of our microglial population was much lower, averaging only 5.4% (figure below), and so is unlikely to drive the microglial subclassification. **As requested, we have now added these feature plots of the subcluster-specific mitochondrial DNA content (Supplemental figure 6d) and nCount (main Figure 6d).**

Functional annotations: Our subcluster annotations were based on recent scRNA seq studies on AD microglia (Sala Frigerio, Friedman, Sierksma), as has been detailed in the Methods and illustrated in Supplemental Figure 6a-c.

General concerns about the quality of the scRNAseq procedure: We hope the preceding clarifications alleviate some of the Reviewer's concern. We are in complete agreement with the Reviewer that great care must be taken in sample processing for scRNAseq, particularly when focusing on microglia, in order to minimize artifactual responses of the cells. Protocols in the literature vary widely and are constantly evolving. In establishing our own procedure, we first tested multiple dissociation and processing strategies published in high impact journals (eg, Harris et al, Cell Stem Cell, 2021; Bottes et al, Nat Neurosci, 2021; Ding et al, Cell Reports, 2020; Hochgerner et al, Nat Neurosci, 2018; Harris et al, PLOS Biol, 2018), finally settling on an optimized and streamlined workflow that significantly reduces the time cells spend *ex vivo* because it removes stressful purification steps found in many other protocols (such as FACS and Percoll steps). As the Reviewer pointed out, there remains a papain dissociation step (as is the case in all of the articles referenced above), which we found necessary for effective dissociation of the adult hippocampus, but we reduced the enzymatic time by 30% versus most other protocols. Importantly, we are confident in the validity of our conclusions as we clearly identified the microglial subclusters that have been reported in other papers, and the gene expression differences we detected in 3xTg microglia where relevant to those seen in other studies using different AD models (Figure 5j).

Multiplexing of samples: We prioritized multiplexing of samples in order to enable all experimental groups to be sequencing in the same run (eliminating batch effects). However, current limitations in the field limit the number of individual samples that can be multiplexed at once. We multiplexed the 4 experimental groups (WT-D, 3xTg-D, WT-S, 3xTg-S), with each group consisting of 10 000 cells pooled together from 4 biological replicates. While cells from the experimental groups can be de-multiplexed to identify their origin, the individual biological replicates all carry the same multiplexing label (cannot be distinguished bio-informatically).

Minor:

The 3xTG AD mouse is never introduced. The three mutations and general course of disease should be described in introduction or results.

>>> **Response:** We have now briefly described how these mice were originally generated and their pathology development in both the Results (lines 98-100) and Methods (lines 744-760) sections.

Fig 2b- the raw annotations and labels are extremely difficult to read. Please simplify for the reader.

>>> **Response:** This has been modified.

Fig3a-b- please define your abbreviations in the legend.

>>> **Response** This has been modified.

Fig 3- I interpreted that all tests underwent two way anova with dunnets? Was 3a a repeated measures anova?

>>> **Response:** Thank you for pointing this out, we had neglected the repeated measures in 3a. **This has now been corrected.**

Figure 4- please clarify hippocampal region where Sholl analysis is performed on the the y-axis of plots, or make the legends less confusing.

>>> **Response:** This has been modified.

Figure 4- how many animals per group are represented in the spine quantifications? Please show means per animal next to the total data distribution, and clarify N's in the figure legend.

>>> **Response:** For spine quantifications, 23 neurons were measured from each experimental group, and these neurons were derived of an N=4 animals per group. We provided means per group, and not per animal, because some animals had relatively fewer neurons (only neurons having a clearly defined dendritic tree that did not overlap with other labelled neurons can be quantified). **The Figure Legend has been modified as requested.**

4i: bulk transcriptomic data on IEGs is not sufficient to make claims about neuronal activity, this should be done in situ with novel exposure followed by c-fos staining, for example.

In addition what do the authors make of the finding that BDNF is so robustly induced by the treatment in WT mice- could suggest alternative mechanisms of spine promotion not related to lipid metabolism per se?

>>> **Response:** Agreed, we did not intent to overstate the link between IEG expression in bulk transcriptomic data and neuronal activity specifically. The text has now been modified as follows (lines: 265-273): “To explore whether the recovery of dendritic spines and dendritic structure was associated with increased markers of cellular activation within the hippocampus, we examined the expression of 18 hallmark immediate-early genes (IEGs) within our bulk RNAseq dataset. IEG expression is rapidly triggered in neuronal and non-neuronal cells in response to diverse signals, and in neurons in particular, IEGs such as *egr-1*, *c-fos*, *Arc* and *BDNF* are dynamically modulated by depolarization [24-27]. Strikingly, 11/18 IEGs were down-regulated in 3xTg-D compared to WT-D mice (2-way ANOVA, Dunnett’s post-hoc test, $p=0.001$), and after SCDi infusion, 3xTg-S mice no longer differed from WT-D mice (Dunnett’s post-hoc test, $p=0.937$) (Fig. 4j).”

Regarding BDNF specifically: Given its pleiotrophic roles in brain development and function, it is indeed interesting that its expression is upregulated by SCDi. We note that BDNF expression can be initiated from multiple promoters, with its promoter IV being activated by neuronal activity (ex. Sakata et al, PNAS, 2009); hence, while the BDNF upregulation might be *contributing* to the observed increases in spines, it might also be a downstream *consequence* of increased synaptic activity.

Figure 5- please provide a legend to clarify the treatment groups (c, j, k), please make the figure legend understandable without text. For example, 5d represents which DEGs? Of the microglial subcluster? Or all cells? For 5 g,h,i, cluster numbers should be consistent throughout, including in the violin plots. The multi letter abbreviations are not helpful, as the reader does not know what they mean, and the interpretations may not be correct. For 5b- it might be more fruitful to focus on subcluster analysis of cd45+ only, since that was the goal of the isolation strategy, and as currently presented, the main take home point of the panel is that the selection works.

>>> **Response:** Thank you for these suggestions, we have accordingly reworked this section in both the figures and associated text.

L284- what is the rationale for using tSNE instead of umap visualization for the subcluster analysis?

>>> **Response:** tSNE and UMAP provide comparable quality of data visualization (local structure). In general, UMAP is more efficient to visualize large cluster differences, while tSNE may offer better visualization of the smaller differences within a cluster (i.e., subclustering) (Kobak and Linderman, Nat Biotech, 39(2):156-157). With our own dataset, we indeed found that tSNE subclustered the microglia in a more clear and visually appealing way than UMAP.

I am somewhat surprised that there are fewer macrophages in the AD model brains – can the authors refer to other literature on the topic that might support this finding?

>>> **Response:** Macrophage data were reported simply to confirm that they were efficiently tagged by the CD45 multiplexing strategy, but we do not consider the absolute number of macrophages to be sufficiently high to interpret changes between experimental groups (a total of between 30 and 170 cells per pooled group of 4 animals). **To avoid confusion, we have now removed discussion of differences between experimental groups for low abundance cell types.**

L173-4- please show the data on swim speed

>>> **Response:** The swim speed data has been added to the Results section (lines 216-219).

Reviewers' Comments:

Reviewer #1:

Remarks to the Author:

Response from Reviewer 1:

This resubmission by Hamilton et al., is now even more compelling, particularly given the significant addition of convincing data in mice using a second, distinct SCD inhibitor. The absence of decreases in monounsaturated fatty acids 16:1 and 18:1 upon SCD inhibition is a little concerning; however, the increase in SCD substrates 16:0 and 18:0 alters the desaturation index as expected and is therefore likely to alter membrane fatty acyl composition accordingly. The authors explain that the chronic nature of their SCDi treatment (as opposed to acute experiments) may explain why 16:0 and 18:0 rose without 16:1 and 18:1 decreasing; other lipid metabolic pathways (e.g., phospholipase activity on membranes) may be compensating.

As acknowledged by the authors, it does not appear possible to definitively distinguish between the reversal of phenotypes being due to restored DEGs or to new DEGs. However, the authors have made a reasonable case for the restored DEGs supporting the phenotype rescue. The authors acknowledge in the text aspects of the data that require follow up or careful interpretation. Hence, the authors have done a careful and thorough job responding to all three reviews, and the paper is now highly worthy of publication. It will be of interest to those in the neurodegenerative field, connecting fatty acid metabolism to microglia and to AD-relevant outcomes and suggesting a novel therapeutic approach for AD.

Reviewer #2:

Remarks to the Author:

General points regarding data in revised manuscript:

- Great addition of other SCD inhibitor! Especially that both suggest "restored" genes may be the most beneficial (as opposed to "new" DEGs following SCDi)
- Good to see saturation changes in vivo; changes seem consistent with expected biological effect of SCDi (increased saturated fatty acids following treatment)
- Interesting to see increase in DHA, given correlation with lowering AD risk
- SCDi microglia experiments demonstrating key role for SCD in microglial MHC-I activation also valuable. Recent work from Yadong Huang lab showed APOE expression correlated with MHC-I in neurons (Zalocusky et al, Nature Neuro 2021); it is interesting to speculate if the same is the case in microglia, and this may be an APOE dependent effect. SCDi rescue may be even more effective in APOE4 carriers, which have particularly disrupted lipid metabolism?

The authors have very significantly improved the manuscript, particularly with addition of the second SCD inhibitor (an experiment recommended/suggested by another reviewer) as well as the detailed lipidomics. While the lipidomics were not done in the region largely analyzed (hippocampus), the authors explained adequately their attempts to gain lipid data from these areas (MALDI, unsuccessful) and reasons for selecting the region they did (SVZ, higher lipid content, etc). The resulting data are very convincing that SCDi is indeed contributing to changes in fatty acid saturation, and that multiple independent SCD inhibitors can achieve the overall benefits.

The text has also been significantly improved, making the experimental steps easier for the reader to follow.

As a minor point: While the SCDi experiments do strongly support the results of the single cell RNAseq data, this is not quite the validation that the reviewer meant in their suggestions. It would still be beneficial to see some of the gene expression changes detailed in the single cell experiment validated in 3xTg treated and untreated tissue, by IHC or RNAscope.

Overall, the revisions have greatly improved the manuscript, and it is recommended for publication.

Reviewer #3:

Remarks to the Author:

The addition of another drug to the experiments is an important improvement to the paper. However, the authors did not revised the synapse data as I requested - means per animal are not shown, therefore it is not possible to evaluate the rigor of this data- given the relatively large p-value calculated from a very inflated number of points, it is not clear whether this data is really meaningful. In addition, they have added an analysis of microglial engulfment that suffers from the same statistical limitation, and the staining for PSD95 is so sparse that it does not appear to have worked. The logical link between the microglial studies and the rest of the paper remains somewhat tenuous, although I would not consider it essential to acceptance of the paper.

REVIEWER COMMENTS

Reviewer #1 (Remarks to the Author):

This resubmission by Hamilton et al., is now even more compelling, particularly given the significant addition of convincing data in mice using a second, distinct SCD inhibitor. The absence of decreases in monounsaturated fatty acids 16:1 and 18:1 upon SCD inhibition is a little concerning; however, the increase in SCD substrates 16:0 and 18:0 alters the desaturation index as expected and is therefore likely to alter membrane fatty acyl composition accordingly. The authors explain that the chronic nature of their SCDi treatment (as opposed to acute experiments) may explain why 16:0 and 18:0 rose without 16:1 and 18:1 decreasing; other lipid metabolic pathways (e.g., phospholipase activity on membranes) may be compensating.

As acknowledged by the authors, it does not appear possible to definitively distinguish between the reversal of phenotypes being due to restored DEGs or to new DEGs. However, the authors have made a reasonable case for the restored DEGs supporting the phenotype rescue. The authors acknowledge in the text aspects of the data that require follow up or careful interpretation. Hence, the authors have done a careful and thorough job responding to all three reviews, and the paper is now highly worthy of publication. It will be of interest to those in the neurodegenerative field, connecting fatty acid metabolism to microglia and to AD-relevant outcomes and suggesting a novel therapeutic approach for AD.

>>>Response: Thank you very much for your review.

Reviewer #2 (Remarks to the Author):

General points regarding data in revised manuscript:

- Great addition of other SCD inhibitor! Especially that both suggest “restored” genes may be the most beneficial (as opposed to “new” DEGs following SCDi)
- Good to see saturation changes *in vivo*; changes seem consistent with expected biological effect of SCDi (increased saturated fatty acids following treatment)
- Interesting to see increase in DHA, given correlation with lowering AD risk
- *Scd*^{-/-} microglia experiments demonstrating key role for *Scd* in microglial MHC-I activation also valuable. Recent work from Yadong Huang lab showed APOE expression correlated with MHC-I in neurons (Zalocusky et al, *Nature Neuro* 2021); it is interesting to speculate if the same is the case in microglia, and this may be an APOE dependent effect. SCDi rescue may be even more effective in APOE4 carriers, which have particularly disrupted lipid metabolism?

The authors have very significantly improved the manuscript, particularly with addition of the second SCD inhibitor (an experiment recommended/suggested by another reviewer) as well as the detailed lipidomics. While the lipidomics were not done in the region largely analyzed (hippocampus), the authors explained adequately their attempts to gain lipid data from these areas (MALDI, unsuccessful) and reasons for selecting the region they did (SVZ, higher lipid content, etc). The resulting data are very convincing that SCDi is indeed contributing to changes in fatty acid saturation, and that multiple independent SCD inhibitors can achieve the overall benefits.

The text has also been significantly improved, making the experimental steps easier for the reader to follow.

As a minor point: While the SCD^{-/-} experiments do strongly support the results of the single cell RNAseq data, this is not quite the validation that the reviewer meant in their suggestions. It would still be beneficial to see some of the gene expression changes detailed in the single cell experiment validated in 3xTg treated and untreated tissue, by IHC or RNAscope.

Overall, the revisions have greatly improved the manuscript, and it is recommended for publication.

>>>Response: Thank you very much for your review and for recommending publication of our study. We understand and agree in principle with the “minor point” about validating gene expression changes observed by scRNAseq. In this regard, we validated our scRNAseq at 2 levels: i) *bio-informatically*, by comparing our microglial DEGs with those reported in other studies (Fig. 5j) and also confirming that we detect previously reported microglial subtypes (Fig. 6a,b), and ii) *functionally*, by showing via qPCR that the main SCDi-modulated MHC-I molecules are indeed affected in *Scd*1-KO microglia (Fig. 5k-n). The reason we did not seek to validate expression changes of individual genes *in situ* is that the shifts in microglial subtypes seen across our experimental groups involved gene expression changes that were i) relatively small in magnitude and ii) differentially altered across microglial subtypes. Thus, since such changes would be technically challenging to even detect (and not overly convincing even if successful), we focused instead on the above bio-informatic and functional validations.

Reviewer #3 (Remarks to the Author):

The addition of another drug to the experiments is an important improvement to the paper. However, the authors did not revised the synapse data as I requested - means per animal are not shown, therefore it is not possible to evaluate the rigor of this data- given the relatively large p-value calculated from a very inflated number of points, it is not clear whether this data is really meaningful. In addition, they have added an analysis of microglial engulfment that suffers from the same statistical limitation, and the staining for PSD95 is so sparse that it does not appear to have worked. The logical link between the microglial studies and the rest of the paper remains somewhat tenuous, although I would not consider it essential to acceptance of the paper.

>>>Response: Thank you again for your review. Regarding the graphs, we apologize for not having revised them previously. We have now re-expressed the aggregate cellular data on a per animal basis as requested, with clear specification of the number of cells per animal that were evaluated in the figure legends (See new Fig. 4 and new Supplemental Fig. 4). Importantly, the data still demonstrate reduced spine numbers in the 3xTg hippocampus, especially in the CA1, that are rescued upon infusion of either the original SCDi (Fig. 4) or the newly added SCDi (Supplemental Fig. 4). A graph of the PSD95/Iba-1 data, re-expressed on a per animal basis, has likewise been added to Fig. 5a (inset) and this shows a similar trend to the aggregate cellular data. Regarding the apparently sparse PSD95 labelling shown in our original Fig. 5b: the expected dense synaptic labelling is in fact seen on our sections, but we had adjusted the brightness to be consistent with the fact that only puncta larger than 0.4 microns were quantified (this was done to avoid quantifying any background labelling). We have now clarified this in the methods, and slightly brightened the image in Fig. 5b to show the smaller puncta that are also present.

(Fig. 5b, slightly brightened to show smaller PSD95 puncta)

Reviewers' Comments:

Reviewer #3:

None